# Large Action Models: From Inception to Implementation

Lu Wang[1]*, Fangkai Yang[1]*, Chaoyun Zhang[1]*, Junting Lu[2], Jiaxu Qian[1],
Shilin He[1], Pu Zhao[1], Bo Qiao[1], Ray Huang[1], Si Qin[1], Qisheng Su[2],
Jiayi Ye[3], Yudi Zhang[4], Jian-Guang Lou[1], Qingwei Lin[1], Saravan Rajmohan[1],
Dongmei Zhang[1], Qi Zhang[1]

[1]**Microsoft**
[2]**Peking University**
[3]**Zhejiang University**
[4]**Eindhoven University of Technology**
{wlu, fangkaiyang, chaoyunzhang}@microsoft.com

**Reviewed on OpenReview:** https://openreview.net/forum?id=3969

## Abstract

As AI continues to advance, there is a growing demand for systems that go beyond language-based assistance and move toward intelligent agents capable of performing real-world actions. This evolution requires the transition from traditional Large Language Models (LLMs), which excel at generating textual responses, to Large Action Models (LAMs), designed for action generation and execution within dynamic environments. Enabled by agent systems, LAMs hold the potential to transform AI from passive language understanding to active task completion, marking a significant milestone in the progression toward artificial general intelligence.

In this paper, we present a comprehensive framework for developing LAMs, offering a systematic approach to their creation, from inception to deployment. We begin with an overview of LAMs, highlighting their unique characteristics and delineating their differences from LLMs. Using a Windows OS-based agent as a case study, we provide a detailed, step-by-step guide on the key stages of LAM development, including data collection, model training, environment integration, grounding, and evaluation. This generalizable workflow can serve as a blueprint for creating functional LAMs in various application domains. We conclude by identifying the current limitations of LAMs and discussing directions for future research and industrial deployment, emphasizing the challenges and opportunities that lie ahead in realizing the full potential of LAMs in real-world applications.

## 1 Introduction

In recent years, large language models (LLMs) have demonstrated remarkable advancements across a range of natural language processing (NLP) tasks (Wei et al., 2022a; Brown, 2020; Yang et al., 2023). These models, often incorporating multiple modalities such as language, vision, and speech, have become foundational in numerous AI-driven applications (Thirunavukarasu et al., 2023; Rubenstein et al., 2023; Wang et al., 2023a; Jiang et al., 2024). Their success is evident in systems like question answering in conversational agents (Ma et al., 2023), code generation in GitHub Copilot (Yetistiren et al., 2023), and improved search capabilities in platforms like Bing (Thomas et al., 2024). The key strengths of LLMs—namely their vast knowledge, ability to support multimodal inputs, and capacity for human-like responses—have propelled them to the forefront of AI research (Minaee et al., 2024). Their capability to generalize via zero-shot learning has further expanded the horizons of what AI systems can achieve, making significant contributions to the productivity of both everyday tasks and specialized professional activities. These innovations mark an important milestone on the path toward artificial general intelligence (AGI) (Feng et al., 2024).

---

*Equal contribution.

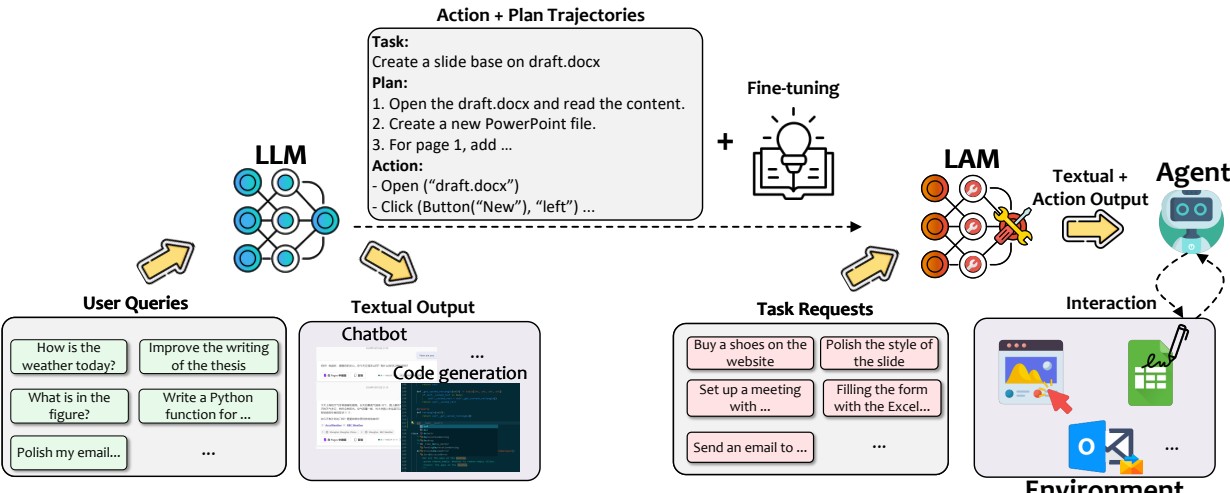

Figure 1: The transition from LLMs to LAMs.

While LLMs excel at generating text, they remain limited in their ability to interact with or manipulate the real world (Wang et al., 2024b). Many real-world tasks require actions beyond language—such as automating software operations or controlling physical devices (Gao et al., 2024). Bridging this gap demands models that not only understand language but can act on it in dynamic environments.

This transition from understanding to execution is non-trivial. Real-world tasks often require multi-step planning, precise action sequencing, and robust adaptation—all areas where current LLMs fall short (Yao et al., 2020; Kalakonda et al., 2023; Valmeekam et al., 2022). LLMs are typically optimized for general-purpose outputs, lacking the task specificity needed for reliable, grounded execution (Ling et al., 2023).

To address these challenges, we propose Large Action Models (LAMs)—LLMs equipped to perform actions in both digital and physical environments (He et al., 2024; Zhang et al., 2025c). LAMs interpret intent, generate structured action plans, and interact with real systems through agents. This represents a critical shift from passive text generation to active, task-completing intelligence.

LAMs are often built upon the foundation of LLMs, but the transition from LLMs to LAMs is neither straightforward nor seamless, as shown in Figure 1. The process of transforming an LLM into a functional LAM involves multiple intricate stages, each requiring substantial effort and expertise. First, it is essential to collect comprehensive datasets that capture user requests, environmental states, and corresponding actions (Deng et al., 2023). These data serve as the basis for training or fine-tuning LLMs to perform actions rather than merely generate text. This stage involves the integration of advanced training techniques that enable the model to understand and execute actions within specific environments (Hong et al., 2024). While GPT-4o demonstrates strong zero-shot capabilities, trained LAMs are better suited for real-world deployment scenarios where latency, cost, and task specialization are critical. In particular, the latency and inference cost associated with GPT-4o can be prohibitive in production settings (especially in high-throughput or interactive environments like GUI automation) making such delays unacceptable. Once the LAM has been trained, it must be incorporated into an agent system that can effectively interact with its environment. This system typically includes components for gathering observations, utilizing tools, maintaining memory, and implementing feedback loops. These components are critical for ensuring that the LAM can not only execute actions but also adapt its behavior based on real-time feedback and evolving situations (Zhang et al., 2025a). The integration of these elements enhances the LAM's capacity to perform tasks autonomously, interact meaningfully with its surroundings, and make decisions that are grounded in the context of its environment.

A critical step in developing Large Action Models (LAMs) is rigorous evaluation (Xie et al., 2024), as LAMs directly impact their environments. This paper presents a practical guide to building LAMs from LLMs, using a GUI agent on Windows OS as a case study. We outline a full pipeline—from data collection to training and grounding—designed for safe, adaptable deployment. Our approach generalizes to other domains and highlights LAMs as a key step toward action-oriented AI, bringing us closer to AGI.

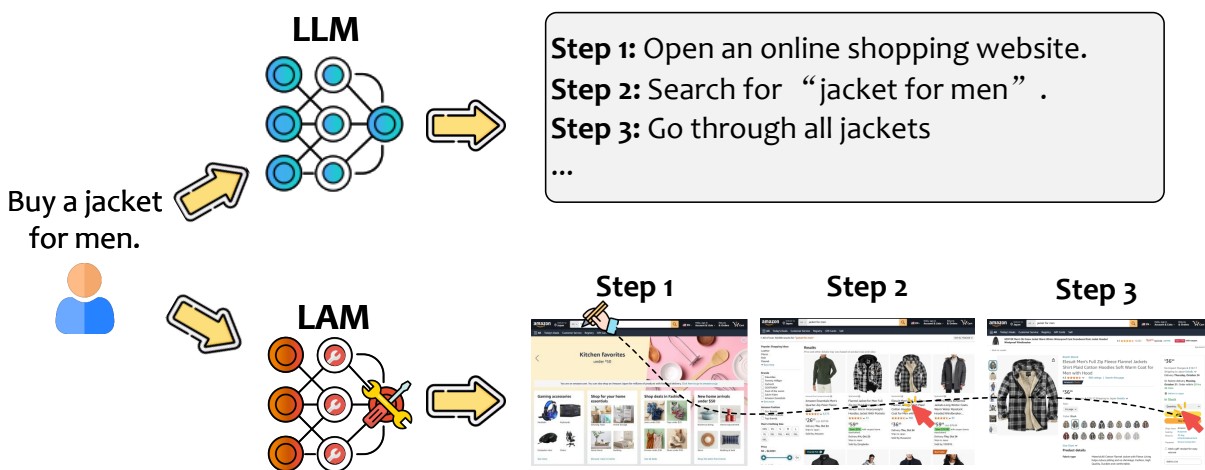

Figure 2: The functional distinction between LLMs and LAMs.

## 2 From LLMs to LAMs

Large Action Models (LAMs) build on the capabilities of Large Language Models (LLMs) but are specifically optimized to execute grounded actions in real-world environments. Unlike LLMs that generate text, LAMs translate user intent into executable operations, allowing them to interact with physical or digital systems to complete tasks (Zeng et al., 2023).

Figure 2 illustrates this functional shift. For instance, an LLM may describe how to buy a jacket, while a LAM can complete the task by navigating the website and processing the checkout steps. This transformation from passive understanding to active execution enables AI agents to deliver practical, task-completing outcomes.

Although often smaller than general-purpose LLMs, LAMs achieve superior performance within specific domains. Their specialization allows for faster inference, lower resource usage, and improved deployment in constrained environments. Training a LAM typically involves domain-specific data, targeted fine-tuning, and integration with agent frameworks that support real-time observation, planning, and tool use (Zhang et al., 2025a).

### 2.1 Key Characteristics of LAMs

**Interpretation of User Intent.** LAMs process abstract or multimodal inputs (such as text, images, or voice) and infer intent through dialogue and contextual reasoning (Chen et al., 2024a; Shah et al., 2023). They convert user requests into high-level task plans. **Action Generation.** LAMs produce executable actions grounded in the current system state and task context. These actions span GUI operations, API calls, robot control, and code generation (Carta et al., 2023).

**Dynamic Planning.** LAMs break down tasks into multi-step sequences and dynamically adapt as the environment changes (Guan et al., 2023; Shinn et al., 2023), ensuring robust execution in dynamic or uncertain settings.

**Domain Specialization.** Trained for specific applications, LAMs incorporate domain-relevant constraints and affordances. Their compact architecture and focused functionality enhance efficiency, accuracy, and deployment feasibility (Cheng et al., 2024).

In summary, LAMs represent a meaningful advancement beyond traditional LLMs. They bridge the gap between passive text-based agents and those capable of performing real-world actions in digital or physical environments.

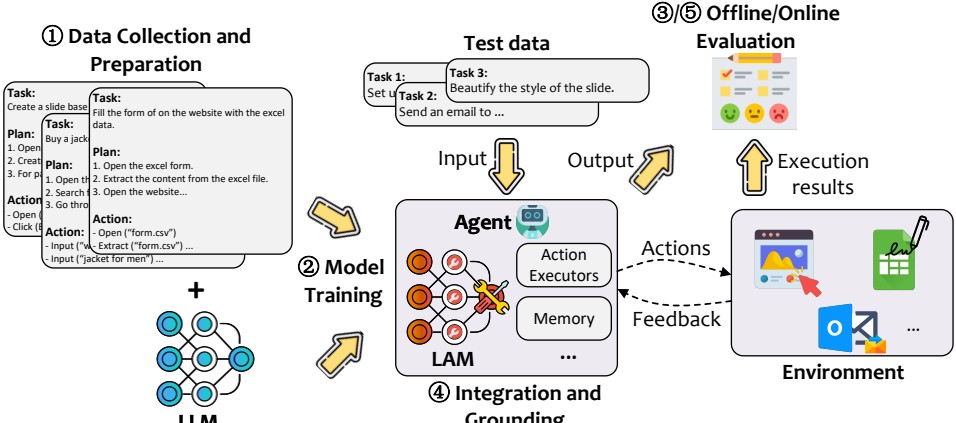

Figure 3: The process pipeline for LAM development and implementation.

## 2.2 From Inception to Implementation

LAMs have the potential to significantly extend the impact of LLMs by enabling tangible interactions with real-world environments. To harness this potential, an LAM must be developed from the ground up and deployed within a real-world application, allowing it to operate effectively in a physical environment. This process involves 5 critical steps, as shown in Figure 3:

1. **Data Collection and Preparation (Section 3)**: The first step involves gathering and curating the necessary data for the specific use case. This includes not only user queries but also environmental context, potential actions, and any other relevant data required to train the LAM effectively. The data must undergo cleaning and pre-processing before it is used for training or fine-tuning a LAM.

2. **Model Training (Section 4)**: Using the prepared data, the next step is to train the LAM. This training process can involve various techniques such as supervised fine-tuning and reinforcement learning to ensure the model can perform the desired actions accurately and efficiently.

3. **Offline Evaluation (Section 5):** After obtaining the LAM, we evaluate its performance using an offline dataset to verify its reliability in a controlled, static environment.

4. **Integration and Grounding (Section 6)**: The LAM is integrated into an agent framework that serves as its operational platform. This involves grounding the model with the ability to interact with external tools, maintain memory, and interface with the environment. By equipping the LAM with these capabilities, it becomes capable of making meaningful impacts in the physical world.

5. **Online Evaluation (Section 7)**: Finally, the performance of the LAM must be rigorously evaluated in the real environment from multiple perspectives, including accuracy, efficiency, and effectiveness in completing tasks. This step is crucial to ensure that the LAM functions as intended and meets the desired operational standards.

Through these steps, LAMs can be effectively developed and deployed to bring LLMs' capabilities into real-world applications, enabling them to interact with and manipulate the physical environment, thereby making a tangible impact.

In the following sections, we use the Windows GUI agent UFO (Zhang et al., 2025a)[1] as a case study to illustrate the process of building a robust LAM from the ground up. This LAM will serve as the core inference engine for UFO, enabling it to autonomously fulfill user requests within the Windows OS environment. While this example focuses on a Windows GUI agent, the outlined steps can be adapted for developing LAMs in other scenarios or for different applications.

---

[1] https://github.com/microsoft/UFO

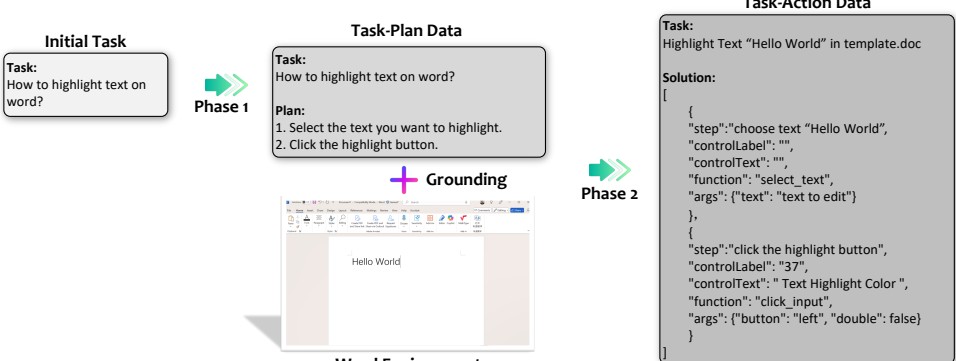

Figure 4: The two-phrase data collection and preparation process.

# 3 Data Collection and Preparation

Data is a cornerstone in training LLMs, where high-quality data significantly enhances their performance (Wang et al., 2023c; Li et al., 2024). Similarly, LAMs require well-prepared, high-quality action-oriented data during the supervised fine-tuning phase. Off-the-shelf LLMs often face challenges when interacting with real-world environments. These difficulties typically arise from either a lack of domain-specific knowledge or the generation of hallucinated outputs that fail to be actionable. To mitigate these issues, we adopt a two-phase data collection approach: *task-plan collection* and *task-action collection*, as shown in Figure 4. Specifically:

1. **Task-Plan Data Collection:** In this phase, we collect data consisting of tasks and their corresponding plans. Tasks are user requests expressed in natural language, while plans are detailed, step-by-step procedures designed to fulfill these requests. For example, a task such as *"How to change the font size in Word?"* would have a corresponding plan outlining the steps required to complete the task. This data is used to fine-tune the model to generate effective plans and improve its high-level reasoning and planning capabilities. However, task-plan data cannot be directly executed in the environment, requiring the following data conversion phase.

2. **Task-Action Data Collection:** In this phase, the task-plan data is converted into task-action data, which includes tasks, plans, and the associated action sequences needed to execute those plans. Tasks and plans are refined to become more concrete and grounded within a specific environment. Action sequences are generated at this stage, such as `select_text(text="hello")` or `click(on=Button("20"), how="left", double=False)`, which represent actionable instructions capable of directly interacting with the environment. This enriched data provides the necessary granularity for training an LAM to perform reliable and accurate task executions in real-world scenarios.

The task-plan data aims at enhancing the model's high-level planning capabilities, allowing it to generate detailed, step-by-step plans based on user requests. Meanwhile, the task-action data focuses on refining the model's ability to execute these plans by converting each planned step into a concrete, executable step or sequence while considering environmental feedback. The data collection and preparation pipeline ensures that the model is capable of both high-level planning and low-level action execution, thereby bridging the gap between natural language plans and executable actions.

In the following sections, we detail the methodologies employed for data collection, pre-processing, and integration of task-plan and task-action data. We illustrate how these steps enable the LLM to LAM transformation.

## 3.1 Task-Plan Data

Figure 5 outlines a multi-step pipeline for collecting and processing task-plan data, essential for training LAMs. The process begins with gathering raw data from diverse sources, including application documentation,

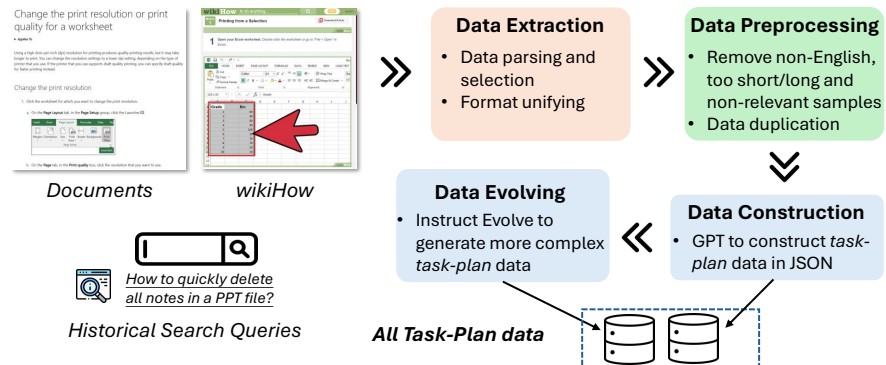

Figure 5: The pipeline to construct the task plan data.

WikiHow, and historical search queries. This is followed by structured pre-processing to ensure that the data is high-quality and relevant to specific tasks.

### 3.1.1 Data Sources

1. **Application Documentation:** Documentation and usage manuals for software applications provide authoritative task descriptions. These resources, maintained by product teams, are considered highly reliable. Relevant documentation, such as M365 documentation[2], is crawled, with outdated or inaccessible pages being filtered out. The HTML content is converted into markdown format, and GPT-4o is used to extract task-plan pairs in the desired structured format.

2. **WikiHow:** WikiHow[3] hosts a wide range of how-to articles, including application-specific operational guides. Webpages related to Windows platform applications are crawled, and GPT-4o extracts task and plan components, ensuring the resulting data aligns with the desired structured format.

3. **Historical Search Queries:** Search engine logs provide insight into real user demands, addressing gaps not covered by formal documentation. From Bing search logs, a 1% sample of queries mentioning application names (*e.g.*, Word, Excel, PowerPoint) from the past year was taken.

### 3.1.2 Data Extraction and Pre-Processing

The first step processes raw data by filtering out non-English, overly short/long, or irrelevant samples (*e.g.*, non-actionable smartphone content). Remaining data is then standardized into a unified format for downstream use.

### 3.1.3 Data Construction

To create structured JSON samples, GPT-4o is employed to extract and format tasks along with their associated plans. For historical search queries, synthetic data is generated to enrich the raw input, addressing the common issue of insufficient context. GPT-4o reformulates these queries into complete, sentence-like user requests, ensuring consistency across all data sources and facilitating effective downstream processing.

The resulting dataset contains structured JSON samples, with each entry including a unique task identifier (task_id), the task description (task), and a step-by-step plan (plan). An example is shown below:

```
1 {"task_id": "word_032",
2  "task": "Add a border to a page in Word",
3  "plan": [
4      1. Go to Design > Page Borders.
```

---

[2]https://learn.microsoft.com/en-us/microsoft-365/?view=o365-worldwide
[3]https://www.wikihow.com/Main-Page

```
5        2. Make selections for how you want the border to look.
6        3. To adjust the distance between the border and the edge of the page,
            select Options. Make your changes and select OK.
7        4. Select OK.
8        ]
9 }
```

With the above process, we initially collected a total of 29,182 task-plan data samples.

### 3.1.4  Data Evolving

With the initial dataset processed, we employ data augmentation techniques to enhance its diversity and complexity. Inspired by WizardLM (Xu et al., 2024) and AgentGen (Hu et al., 2025), we use GPT-4o to evolve the raw task to generate new task-plan pairs, improving the model's ability to follow instructions and handle more complex tasks.

The data evolving process generates new tasks from the original ones by introducing additional complexity, constraints, or steps while preserving relevance. The guidelines for task evolution are as follows:

– The evolved task must be executable step-by-step on a Windows OS or application.

– The evolved task should include additional requirements, increasing its complexity without exceeding 20 extra words.

– The evolved task must remain concise and related to the original task.

For each evolved task, GPT-4o generates a corresponding plan adhering to the following guidelines:

– The plan must provide correct and actionable steps for Windows environments or applications.

– The plan should be concise and highlight critical action objects using bold emphasis.

This augmentation process results in a richer dataset where tasks become progressively more challenging, and plans incorporate domain-specific knowledge. For example:

*Raw task*: Create a drop-down list in Excel for Office 365.

*Evolved Task*: Create a dependent drop-down list in Excel for Office 365, where selecting an item from the first list filters options in the second list.

*Evolved Plan*:

– Prepare your data by organizing it into two columns. The first column contains items for the primary drop-down list, and the second column contains items for the dependent list.

– Name your ranges for the first and second lists.

– Create the primary drop-down list using Data Validation.

– Use the INDIRECT function to create the dependent drop-down list linked to the first selection.

– . . .

Using data augmentation, we increased the original task-plan dataset by 150%, generating a larger pool of samples. This augmentation significantly enhances the diversity and complexity of the dataset, allowing the model to learn from a broader range of scenarios and develop robust planning capabilities. The augmented data introduces more challenging tasks and detailed plans, further enriching the training process and enabling the LAM to handle complex real-world applications effectively.

### 3.2 Task-Action Data

The task-plan data collected in the previous stage provides high-level, step-by-step plans for resolving user-requested tasks, serving as general guidelines. However, these plans are textual and not directly executable in a real-world environment. For instance, a task-plan data sample for the task "Highlight text in document" outlines the necessary steps but does not translate into actionable instructions for interacting with the application's GUI. This gap highlights the need for actionable task-action data to bridge the divide between planning and execution. To enable LAMs to produce actionable outputs, we generate task-action data derived from the previously collected task-plan data. Task-action data captures the granular interactions required to complete a task in the application environment, including GUI navigation, button clicks, and responding to environmental feedback.

Traditional approaches for action data collection often involve manual or agent-based annotation for each task, which is both costly and labor-intensive. To address these limitations, we propose an efficient, fully automated, and low-cost pipeline that leverages LLMs and real-world application interactions. This pipeline consists of four stages, as depicted in Figure 6: **Instantiation**, **Execution**, **Evaluation**, and **Post-Processing**. Specifically,

1. **Instantiation:** In this stage, the task-plan data is transformed into an executable trajectory. Using an LLM, each task is instantiated with specific operational objects, and related high-level plan is instantiated into a concrete sequence of actions that can be directly executed in the application environment.

2. **Execution:** The instantiated trajectory is then executed within the real-world application environment. During this stage, the system interacts with the application's GUI to carry out the specified actions. For example, the instantiated trajectory for highlighting text would involve selecting the appropriate text, navigating to the highlight tool, and applying the highlight. The result of this execution is the captured executed trajectory, including any feedback or environmental changes observed during the process.

3. **Evaluation:** Once the execution is complete, the trajectory is evaluated for correctness using an LLM. The evaluation stage verifies whether the executed trajectory successfully accomplishes the intended task. This involves comparing the observed outcomes with the expected results outlined in the task-plan data. Tasks that fail to meet the criteria are flagged for review, while successful executions are retained for further processing.

4. **Post-Processing:** In the final stage, successful task-action trajectories undergo post-processing to ensure consistency, completeness, and readiness for training. This includes refining the data format, ensuring compatibility with the training pipeline, and annotating the data with relevant metadata (*e.g.*, task IDs, execution time, and step-by-step feedback). The post-processed task-action data is then added to the training dataset, enabling the LAM to learn from real-world interactions.

The pipeline minimizes human intervention and reduces the number of LLM calls required, significantly improving scalability and efficiency.

#### 3.2.1 Instantiation

The task-plan data are primarily collected from help documents or public websites, creating a gap between the generalized task-plan data and the specific requirements needed for execution within a particular environment. A common issue is the lack of specificity. For instance, the task *"highlight text in document"* does not specify actionable objects, such as *"which text"* or *"which document"*. This lack of detail poses significant challenges in executing tasks within real-world applications.

To address this problem, we instantiate the task-plan data to impute target objects and related functions. First, we prepare template Word files to serve as specific targets for the actions. These template files include various Word components such as paragraphs, tables, and figures. Each template file is accompanied by a description indicating its content, providing context for grounding actions. Several sample template files can be found in Appendix A.

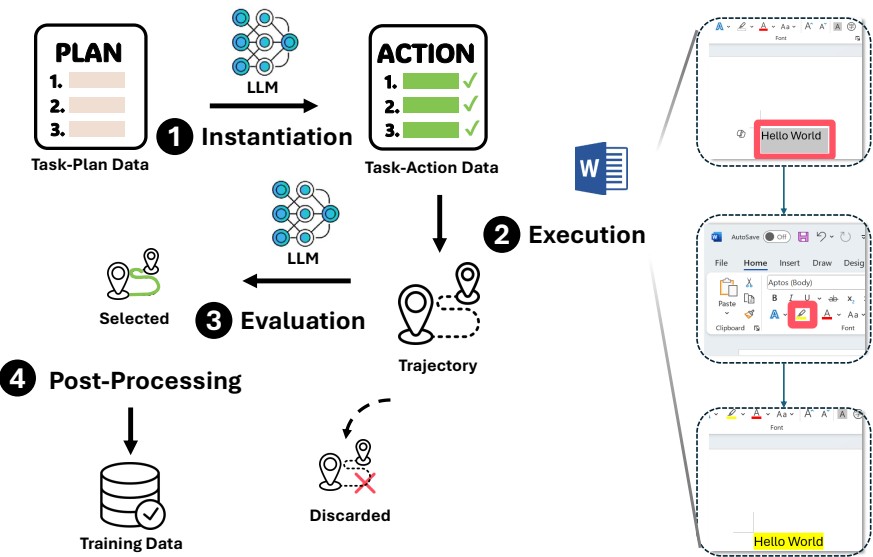

Figure 6: The pipeline of task-action data conversion and collection.

Given a task-plan data sample, the task description is matched with the template file descriptions to select an appropriate template file as the target for actions. GPT-4 is then prompted to instantiate the task-plan with target objects present in the selected template file (detailed prompts can be found in Appendix B.1). Simultaneously, we filter relevant functions from the available function pool using the task description, allowing the instantiation process to populate the task-action data with specific functions and their input parameters.

As a result of this process, the task description becomes more concrete and grounded in a specific environment, while the corresponding action sequences needed to complete the task are generated. Figure 7 provides an example of the instantiation process. Notably, the task-action data is not directly generated with GPT-4 due to the risk of hallucinations. Instead, instantiating grounded task-plan data ensures the generation of more reliable and faithful step-by-step actions.

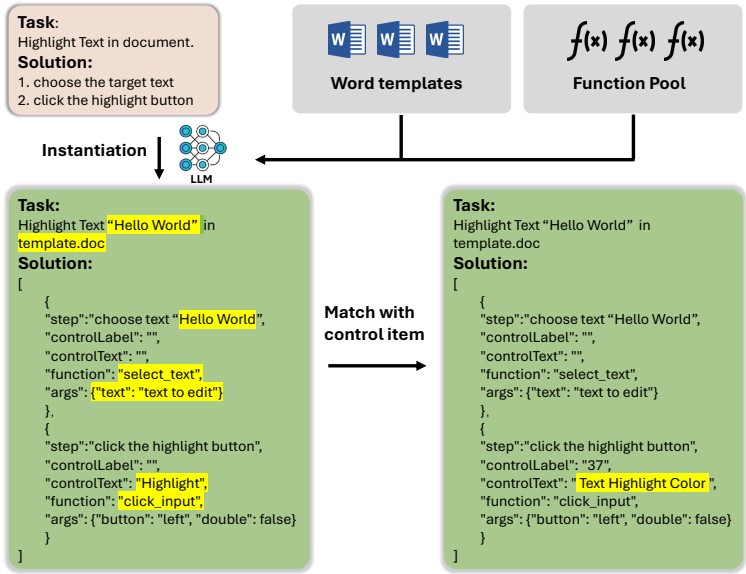

Figure 7: An example of task instantiation.

### 3.2.2 Execution

To ensure that the steps in the instantiated task-plan data are accurate and truly actionable, the execution stage verifies the action sequence by matching control items with the real application environment and performing the specified actions. This process validates the task-action data, ensuring its correctness and compatibility with the application GUI.

For instance, as shown in Figure 7, the control item *"Text Highlight Color"* with its associated control label is retrieved using the action text *"Highlight"* from the control item pool. The corresponding task-action data is then executed in the application without further intervention from the LLM. During execution, if an error occurs (*e.g.*, a mismatch between the predicted control item and the actual environment), the instantiated task is discarded. Conversely, if all actions in the task execute successfully, the action-validated task is forwarded to the evaluation stage described in the following section. Additionally, screenshots of the application environment are captured after each step in the execution process, forming a detailed trajectory to assist in subsequent evaluation.

It is important to note that the instantiated task-action data is not guaranteed to be valid. Since the data is generated through a single GPT-4 call based on task-plan data, it lacks the step-by-step refinement that might be necessary for certain tasks. In some cases, execution results from previous steps are required to instantiate subsequent steps accurately. In such scenarios, the one-call instantiated task-action data may fail in validation and is removed from the dataset. This execution stage bridges the gap between planning and action, ensuring that task-action data is actionable, robust, and aligned with real-world application requirements.

### 3.2.3 Evaluation

Even if a task-action trajectory executes without error, additional validation is needed to ensure it fulfills the original task intent. Some plans may yield executable but incorrect behaviors. To evaluate correctness, we use: (1) the full action sequence, (2) before/after screenshots, and (3) environmental changes (*e.g.*, comparing `.xml` state files in Word[4]). We prompt GPT-4o to assess whether the outcome matches the task description, assigning a `task-complete` key ("yes", "no", or "unsure"). Only tasks marked `"yes"` are retained for training, ensuring data quality. Full prompt details are in Appendix B.2.

### 3.2.4 Post-Processing

As noted in Section 3.2.2, a trajectory was recorded during the execution process. This trajectory includes: (1) Screenshots captured at each step. (2) Environment states before and after each action. (3) Plans and corresponding actions for every step.

During the post-processing stage, these trajectories are combined with the original task requests to generate synthetic step-wise training data. The resulting data format uses the task request as input and LAM's plan and actions as output. This structured format is critical for training LAMs to map task requests to actionable sequences effectively. The detailed template for the data format can be found in Appendix C.

## 4 Model Training

Our objective is to develop an LAM from zero human-labeled data that can map user inputs to appropriate plans and executable actions, ultimately enabling complex task completion. To achieve this, we adopt a staged training strategy consisting of four phases, each building upon the previous one. As illustrated in Figure 8, these phases guide the model from learning structured task plans, to imitating expert demonstrations, to self-boosting from its own successes, and finally leveraging reward-based optimization. Throughout these stages, the model progressively evolves from $LAM^1$ to $LAM^4$.

At a high level, **Phase 1: Task-Plan Pretraining** provides a strong foundation by teaching the model to generate coherent, step-by-step plans for various tasks. **Phase 2: Learning from Experts** then introduces

---

[4]These files represent the underlying document structure.

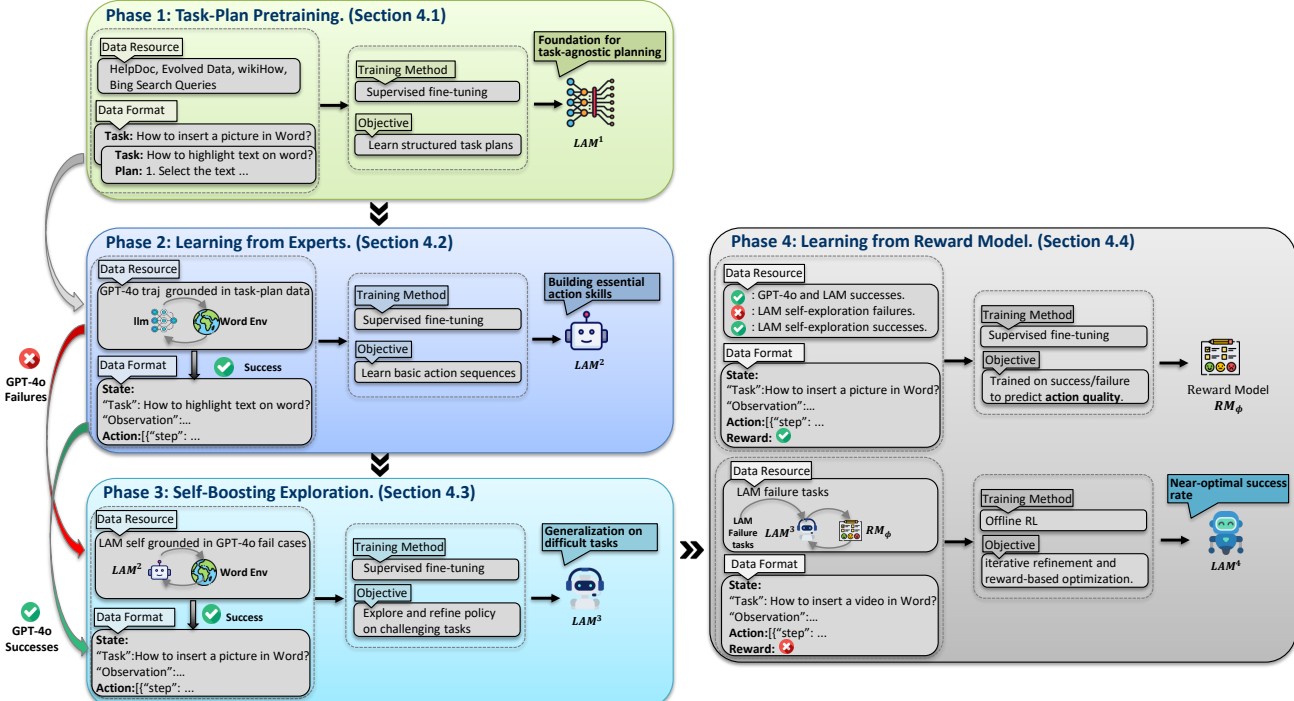

Figure 8: The overview of LAM training pipeline.

action trajectories labeled by GPT-4o, enabling $LAM^2$ to align its plan generation with actionable steps. However, relying solely on expert successes limits diversity and adaptability. To address this, **Phase 3: Self-Boosting Exploration** encourages the model to tackle tasks that even GPT-4o failed to solve, autonomously generating new success cases and evolving into $LAM^3$. Finally, **Phase 4: Learning from a Reward Model** incorporates reinforcement learning (RL) principles, allowing $LAM^4$ to learn from both successes and failures, refining its decision-making in complex, previously unseen scenarios. Table 1 summarizes the data used in each phase. Each phase uses different training objectives, namely *(i)* task-plan pretraining (phase 1) and *(ii)* decision-making training (phase 2-4), as detailed in Appendix E.

Table 1: Training data summary for each phase of LAM training.

| Model | Data Type | Data Source | Input → Output | Data Size |
|---|---|---|---|---|
| $LAM^1$ | Task-Plan Pairs | Application documentation, WikiHow, evolved data historical search queries | $t_i \rightarrow P_i$ | 76,672 tasks |
| $LAM^2$ | Task-Action Trajectories | GPT-4o | $s_t \rightarrow a_t$ | 2,192 trajectories |
| $LAM^3$ | Task-Action Trajectories | $LAM^2$ + GPT-4o | $s_t \rightarrow a_t$ | 2,688 trajectories |
| $LAM^4$ | Task-Action-Reward Trajectories | RM + $LAM^3$ | $(s_t, r_t) \rightarrow a_t$ | 1,788 trajectories |
| Reward Model | Task-Action-Reward Trajectories | GPT-4o + $LAM^3$ | $(s_t, a_t) \rightarrow r_t$ | 4,476 trajectories |

## 4.1 Phase 1: Task-Plan Pretraining

The initial stage focuses on imparting a broad understanding of how tasks can be decomposed into logical steps. We start with Mistral-7B (Jiang et al., 2023) as the base model. A total of **76,672** task-plan pairs $(t_i, P_i)$ are collected from various sources, including application help documentation, WikiHow, and historical search queries. Of these, 29,182 pairs are sourced directly, while 47,490 are generated via data evolution techniques (as described in Section 3.1.4), enriching the dataset with more complex and diverse tasks.

In this phase, $\text{LAM}^1$ is trained via supervised fine-tuning (SFT) (Deng et al., 2023) to predict the correct plan sequence $P_i$ for a given task $t_i$:

$$\mathcal{L}_{\text{SFT}}(\text{LAM}_\theta^1) = \frac{1}{N} \sum_{i=1}^{N} \mathcal{L}_{\text{CE}}(P_i^{\text{pred}}, P_i^{\text{true}}).$$

Here, $\mathcal{L}_{\text{CE}}$ denotes the cross-entropy loss, and $N$ is the number of tasks. Although no actions are generated at this stage, $\text{LAM}^1$ gains a robust planning capability. This knowledge will prove critical in guiding the model's action execution in later phases, ensuring that the agent understands the logical structure of tasks before attempting to perform them.

### 4.2 Phase 2: Learning from Experts

While $\text{LAM}^1$ can produce structured plans, it lacks the ability to execute them. In Phase 2, we introduce expert-labeled task-action trajectories from GPT-4o (Section 3.2) to teach the model how to perform actions. The illustrative application in this paper is the Microsoft Word environment, where we have **2,192** successful expert trajectories. Each trajectory consists of a sequence of state-action pairs $(s_t, a_t)$, representing observed UI states and the corresponding actions to progress the task.

We split these 2,192 trajectories into a training set of 1,757 and a test set of 435 trajectories, providing a total of 3,959 steps for training. By imitation learning $\text{LAM}^1$ on these successful action sequences, we obtain $\text{LAM}^2$. The objective is to minimize:

$$\mathcal{L}_{\text{SFT}}(\text{LAM}_\theta^2) = \frac{1}{N} \sum_{i=1}^{N} \sum_{t=1}^{T_i} \mathcal{L}_{\text{CE}}(\text{LAM}_\theta^2(s_t), a_t),$$

where $N$ is the number of trajectories and $T_i$ is the number of steps in trajectory $i$. By imitating the expert's policy, $\text{LAM}^2$ transforms from a passive planner into a model capable of executing actions aligned with its plans, grounding its reasoning in the real application environment.

### 4.3 Phase 3: Self-Boosting Exploration

Up to Phase 2, $\text{LAM}^2$ only learns from successful trajectories provided by GPT-4o. This limits diversity and adaptability, as the model never sees how to handle situations that even GPT-4o could not deal with. To overcome this limitation, Phase 3 introduces self-boosting exploration.

Here, we revisit failed GPT-4o trajectories, *i.e.*, tasks that GPT-4o did not complete successfully, and let $\text{LAM}^2$ attempt them. Using the ReAct mechanism (Yao et al., 2023; Shinn et al., 2023), $\text{LAM}^2$ interacts with the environment and tries alternative strategies for these challenging tasks. To construct meaningful task trajectories, the GUI agent uses the ReAct mechanism by interleaving observation, reasoning, and action. At each step, the agent perceives the current GUI state, reasons about the next move, executes the corresponding action, and records the updated environment, action taken, and reasoning trace. These step-by-step interactions are then aggregated to form coherent trajectories for training a LAM.

From these attempts, we sampled 2284 GPT-4o failed tasks and then collect **496** newly successful trajectories generated by $\text{LAM}^2$ itself. These self-labeled successes, combined with the original 2,192 GPT-4o successes, form an augmented dataset.

We then fine-tune $\text{LAM}^2$ on this enriched data, yielding $\text{LAM}^3$:

$$\mathcal{L}_{\text{SFT}}(\text{LAM}_\theta^3) = \frac{1}{N} \sum_{i=1}^{N} \sum_{t=1}^{T_i} \mathcal{L}_{\text{CE}}(\text{LAM}_\theta^3(s_t), a_t).$$

This self-boosting step allows the model to learn from its own newly discovered solutions, overcoming previous limitations and improving adaptability. By leveraging planning knowledge from Phase 1 and expert strategies from Phase 2, $\text{LAM}^3$ becomes more resourceful, even in scenarios with sparse or absent expert guidance.

### 4.4 Phase 4: Learning from a Reward Model

Despite the improvements, Phases 1–3 focus on successes or expert-like behavior. They offer limited insights into intermediate decision quality and fail to exploit learning opportunities presented by failed attempts. In Phase 4, we integrate reinforcement learning (RL) to address these shortcomings (Kasneci et al., 2023).

To this end, we adopt a two-stage approach: first, we train a reward model, and then we further fine-tune the LAM using this model. The reward model (RM) is built using $\text{LAM}^3$ as the base model, with an additional output layer added to produce scalar values representing the quality of actions. Using the trained RM, we fine-tune $\text{LAM}^4$ in an offline RL setting. Here, the model refines its policy without additional environmental interactions, leveraging previously collected trajectories to learn from failures and improve action selection.

#### 4.4.1 Reward Model Training

First, we train a reward model (RM) on both $\text{LAM}^3$'s successful (496) and failed (1788) trajectories and GPT-4o's successful trajectories (2192) gathered in previous phases. All steps in successful trajectories are assigned a reward of $+1$, and all steps in failed trajectories a reward of $-1$. This uniform, binary labeling of outcomes ensures the RM consistently captures overall trajectory quality. Formally:

$$r_t = \text{RM}(s_t, a_t; \phi),$$

where $\phi$ presents the RM parameters, and $r_t \in \{+1, -1\}$ is the assigned reward. The RM is trained via mean squared error (MSE) to approximate these ground-truth rewards.

The training dataset for the RM includes both failed and successful task-action trajectories generated by $\text{LAM}^3$, as well as the successful trajectories from the collected task-action data. All steps in successful trajectories receive a reward of $+1$, while every step in failed trajectories is assigned a reward of $-1$. This uniform labeling strategy ensures that the RM consistently reflects overall trajectory quality and effectively guides policy optimization.

#### 4.4.2 Optimizing with Offline PPO

Armed with the RM to evaluate intermediate actions, we fine-tune $\text{LAM}^4$ via offline PPO (Schulman et al., 2017). This stage focuses on the 1,788 failure trajectories collected during Phase 3, providing a unique opportunity to learn from mistakes. The training objective of PPO is:

$$\mathcal{L}_{\text{PPO}}(\text{LAM}^4_\theta) = \frac{1}{N} \sum_{i=1}^{N} \sum_{t=1}^{T_i} \min\left( \frac{\text{LAM}^4_\theta(a_t|s_t)}{\text{LAM}^4_{\theta_{\text{old}}}(a_t|s_t)} \hat{A}_t, \quad \text{clip}\left( \frac{\text{LAM}^4_\theta(a_t|s_t)}{\text{LAM}^4_{\theta_{\text{old}}}(a_t|s_t)}, 1-\epsilon, 1+\epsilon \right) \hat{A}_t \right),$$

where $\hat{A}_t$ denotes the advantage derived from RM-generated rewards, and $\epsilon$ is a clipping parameter to ensure stable updates.

By incorporating signals from both successes and failures, $\text{LAM}^4$ gains a deeper understanding of action quality. This RL-based fine-tuning helps the model generalize to complex, previously unseen scenarios, ensuring more robust and reliable decision-making. Our pipeline is fundamentally domain-agnostic and consists of: (1) Data Collection & Task-Plan Pretraining, (2) Imitation Learning from Expert Trajectories, (3) Self-Boosting Exploration, and (4) Reward-Model Guided Refinement.

However, to deploy in a new domain, the environment must support the execution, observation, and validation of action trajectories (e.g., via APIs, simulators, or instrumentation). With these capabilities in place, the same four-phase structure can be applied: (1) **Video Games:** Discrete controls (e.g., button presses, joystick directions) can be collected through game APIs or screen-state instrumentation. (2) **Robotics:** The same stages apply, but the model must predict continuous motor commands or trajectories (e.g., using a different decoder head). Simulation platforms (e.g., MuJoCo, Gazebo) facilitate low-risk and scalable data collection.

Table 2: Offline performance comparison across different models and metrics on decision making.

| Metric | LAM[1] | LAM[2] | LAM[3] | LAM[4] | GPT-4o | GPT-4o Mini | DeepSeek R1 | O1-mini |
|---|---|---|---|---|---|---|---|---|
| Object Acc (%) | 39.4 | 85.6 | 87.4 | **87.8** | 73.2 | 68.4 | 63.4 | 71.8 |
| Operation Acc (%) | 59.9 | 97.3 | 97.7 | **97.7** | 94.2 | 89.6 | 87.8 | 88.5 |
| Status Acc (%) | 32.7 | 97.8 | 98.2 | **99.0** | 65.3 | 60.2 | 68.2 | 59.4 |
| Step Success Rate (SSR) (%) | 33.0 | 83.6 | 85.9 | **86.2** | 68.8 | 62.7 | 59.4 | 69.3 |
| Task Success Rate (TSR) (%) | 35.6 | 76.8 | 79.3 | **81.2** | 67.2 | 61.2 | 55.2 | 64.6 |

# 5 Offline Evaluations

The offline evaluation results of **Task-Plan Pretraining Results (Phase 1)** and **Task-Action Results (Phases 2–4)** will be presented in this section. Offline evaluation allows us to systematically assess the performance of LAM[1] and subsequent phases (LAM[2], LAM[3], and LAM[4]) without interacting with the environment. This setup effectively provides a controlled and reproducible framework for comparing task success rates, precision, and recall metrics across models.

## 5.1 Experiment Setup

### 5.1.1 Data Preparation

We took several rigorous steps to ensure that the evaluation set is both held-out and semantically distinct from the training set. (1) **Redundancy Removal Before Splitting:** Prior to partitioning, we applied semantic deduplication to eliminate redundant or highly similar tasks. This ensured that no rephrased or paraphrased version of a training task appears in the test set. (2) **Random Sampling After Cleaning:** The test set was then randomly sampled from the cleaned data. These test tasks were never seen during training or used for model selection. (3) **Semantic Similarity Validation:** To validate the distinctiveness of the test set, we followed (Wang et al., 2023b) by computing ROUGE-L similarity between training and test sets. We observed an average ROUGE-L score of 0.27, indicating that the test set is semantically distinct and suitable for evaluating generalization. For details on our semantic deduplication process and dataset splitting strategy, please refer to Appendix F.

### 5.1.2 Evaluation Metrics

To evaluate agent performance in task execution, we adopt five metrics: Object Accuracy, Operation Accuracy, Status Accuracy, Step Success Rate (**SSR**), and Task Success Rate (**TSR**). Object Accuracy measures whether the agent selects the correct UI element. Operation Accuracy checks if the predicted action (e.g., Click, Type) matches the ground truth. Status Accuracy evaluates whether the agent correctly determines task completion status. Step Success Rate (**SSR**) considers a step successful only if object, operation, and status are all correct. Task Success Rate (**TSR**) requires all steps in a task to be correct for the task to count as successful.

While task success rate (TSR) is our primary metric, we include step success rate (SSR) as a complementary diagnostic tool. In GUI-based environments like Word, most tasks follow highly constrained and canonical trajectories, making step-level precision both meaningful and necessary. Although multiple trajectories may theoretically complete a task, in practice, deviations often lead to errors or suboptimal behavior. Thus, LAM's higher SSR reflects not just closer alignment with reference trajectories but greater consistency, planning quality, and robustness. Also, SSR aligns with standard agent benchmarks (e.g., Mind2Web (Deng et al., 2023), AITW (Rawles et al., 2023)). LAM[2] builds on LAM[1], which learns structured task decomposition from curated plans—knowledge not explicitly present in GPT-4o. Combined with domain-specific fine-tuning, LAM[2] integrates planning priors and labeled actions to achieve more grounded execution. Unlike GPT-4o's zero-shot approach, LAM[2] is adapted to the task interface, reducing hallucinations and improving consistency.

### 5.1.3   Performance on Decision Making

Table 2 presents results on 435 Word tasks, showing that our four-phase LAM framework yields cumulative performance gains. LAM$^4$ achieves a TSR of **81.2**%, outperforming GPT-4o (67.2%), GPT-4o-mini (61.2%), DeepSeek R1 (55.2%), and o1-mini (64.6%). These improvements arise from a stepwise training strategy: task-plan pretraining (LAM$^1$), GPT-4o-labeled imitation learning (LAM$^2$), self-boosting on GPT-4o failure cases (LAM$^3$), and reward-guided fine-tuning (LAM$^4$).

Despite using GPT-4o-labeled data in early phases, LAM surpasses GPT-4o by learning from its limitations and refining its decision-making through domain-specific supervision. The ReAct mechanism supports this by enabling LAM to generate new successful trajectories for harder tasks. Across key metrics including step success rate (86.2%), object accuracy (87.8%), and status accuracy (99.0%), LAM$^4$ consistently outperforms all baselines. These results highlight the effectiveness of our targeted pipeline in producing robust and reliable action models for real-world applications. All updated comparisons have been incorporated into the main text.

As for costs, GPT-4o was used for approximately 50K API calls during data instantiation and validation, incurring a total cost of  2.5$K$ (at 0.05 per 1K tokens), compared to an estimated $> 20K$ for equivalent manual annotation. Inference costs are minimal ($< 0.01$ per task).

### 5.2   Failure Cases and Error Patterns in LAM

While LAM achieves high performance across most tasks, we identify several recurring error patterns that reveal areas for further refinement.1. Early Termination Before Task Completion LAM may halt execution prematurely, mistaking intermediate steps for task completion. *Example: In a task requiring word count calculation, LAM correctly opens the "Review" tab but fails to click on "Word Count," stopping short of the final step.* 2. Skipping Required Preconditions LAM may attempt to execute goal-directed actions without verifying or setting required environmental conditions. *Example: When instructed to calculate word count for a document in A4 format with font size 11, LAM skips the font and layout verification steps and proceeds directly to "Word Count."* 3. Incorrect or Ambiguous UI Element Selection LAM sometimes selects visually similar but incorrect controls due to limited UI disambiguation. *Example: In an Excel autofill task, LAM selects the "Editing" tab instead of directly interacting with the target cells.*

These failure cases reveal three dominant error patterns in LAM's current behavior: (1) premature task termination due to overconfident stopping, (2) insufficient attention to task preconditions, and (3) UI disambiguation failures when interacting with similar elements. Addressing these issues will require improvements in planning, state verification, and more robust grounding to UI context. These kinds of problems we will consider in the future work.

## 6   Integration and Grounding

### 6.1   LAM Agent In a Nutshell

In UFO, the LAM serves as the inference engine within the AppAgent, enabling efficient and accurate task completion. Figure 9 illustrates the architecture of the AppAgent. UFO, equipped with LAMs, is designed for interactive engagement with Windows applications. For simplicity, we focus on automating tasks within Microsoft Word, a widely used productivity tool with a sophisticated GUI and diverse functionalities, making it an ideal testbed for training and evaluating LAM.

During each inference step, the agent collects critical contextual information from the application environment, which is then passed to the LAM for decision-making. The LAM performs planning, orchestrates actions, and infers the necessary steps to fulfill the user request. These inferred actions are grounded in the environment by mapping them to predefined tools and function calls used by the agent, such as mouse clicks, keyboard inputs, or API calls. This process iterates, with LAM continuously adjusting its plan based on real-time feedback from the environment, until the task is completed. Additionally, the agent maintains a memory that logs historical actions and plans, providing essential context for the LAM to make more informed and

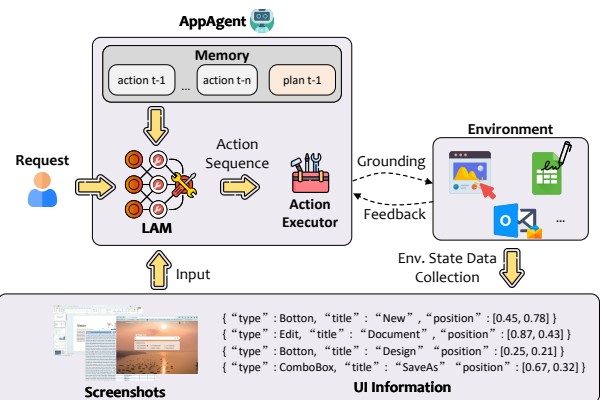

Figure 9: The overall architecture of the AppAgent employed in UFO.

adaptive decisions as the task progresses. This integration ensures that UFO can efficiently manage and complete complex, real-world tasks in Windows environments.

## 6.2 Environment

The UFO agent leverages the LAM to interact with applications in the Windows environment. At each decision step, UFO employs the UI Automation (UIA) API (Dinh et al., 2018) to inspect all actionable controls within the target Windows application, retrieving contextual information for each control[5]. This information is passed to the LAM for control selection and action inference. The control data is structured as a list of dictionaries, where each control is assigned a numerical index (as a label), along with its title and control type, allowing the LAM to make informed decisions regarding control selection and the corresponding action. This input format mirrors the structure used during offline data collection for consistency in training and execution.

## 6.3 LAM Inference

Using the environmental observations of application control information, UFO constructs prompts in the same format as the offline training data, using planning and thought generation techniques (Wei et al., 2022b; Ding et al., 2024) to enable LAM to make reliable inferences about the appropriate controls and operations to invoke. These inferences target the controls detected by the UIA, where each control is selected from a predefined list. The function calls inferred by LAM are limited to pre-defined operations, such as mouse and keyboard actions, as well as APIs specific to Word-related tasks. Once inferred, these operations are parsed and executed within the environment.

## 6.4 Action Execution

UFO employs a control interactor to ground the action strings generated by LAMs, translating them into tangible impacts within the target application. Each action typically consists of two key components: **Control Element:** This refers to the specific UI control within the application that will receive the action, such as a button, text box, or scroll bar. **Function Call:** This represents the operation to be performed on the control element, such as a mouse click, keyboard input, or invocation of native APIs. By combining the control element and its associated function call, UFO executes the inferred actions within the application.

---

[5]UIA is the native Windows OS APIs used to detect actionable controls and provide their metadata, such as names and locations. For other platforms, UIA can be replaced by vision-based detectors that analyze screenshots or by utilizing alternative accessibility APIs.

## 6.5 Memory

UFO maintains additional information in its memory to assist LAMs in making more informed and accurate decisions. This memory includes: (1) **Historical Actions:** A log of action trajectories and their execution results from the initial step onwards. This helps LAM understand the current system state and aids in exploring the next steps based on prior actions. (2) **Previous Plan:** The textual planning for future actions, generated by LAM in the previous step. This serves as a reference for guiding the current and future actions, ensuring consistency across steps.

# 7 Online Evaluations

With the integration of the Windows GUI agent UFO, we evaluate the performance of the LAM in real-world environments. The evaluation process and results are detailed in the following subsections.

## 7.1 Testing Dataset

The online performance of LAM is evaluated on the same set of 435 test requests used during LAM training. The testing environments, specifically the Word document templates corresponding to each task, are also maintained as identical to the training setup to ensure consistency and comparability.

Table 3: Performance comparison of LAM and baseline models across metrics.

| Metric | Text-only | | | Text + Visual | |
|---|---|---|---|---|---|
| | LAM | GPT-4o + UFO | GPT-4o Mini + UFO | GPT-4o + UFO | GPT-4o Mini + UFO |
| Task Success Rate (%) | 71.0 | 63.0 | 57.8 | 75.5 | 66.7 |
| Task Completion Time (s) | 30.42 | 86.42 | 35.24 | 96.48 | 46.21 |
| Task Completion Steps | 5.62 | 6.73 | 5.99 | 4.98 | 6.34 |
| Average Step Latency (s) | 5.41 | 12.84 | 5.88 | 19.36 | 7.29 |

## 7.2 Implementation

Our LAM was deployed on a virtual machine (VM) configured as NC24s v3. The VM is equipped with 24 virtual cores (vCPUs), 448 GB of memory, and two NVIDIA Tesla V100 GPUs, each with 16 GB of memory, to support efficient inference. This computational setup was designed to meet the demanding requirements of LAM's inference processes effectively.

The UFO agent operates on six VMs running in parallel using Azure Dedicated Host[6] to accelerate the testing process. Each VM is equipped with a 15-core Intel(R) Xeon(R) Platinum 8370C CPU @ 2.80GHz, 64GB of RAM, and runs Windows 11 Enterprise version 23H2. Microsoft applications, such as Word and Excel, are installed on version 2410. GUI control is facilitated through the MSTSC tool[7]. This setup ensures a consistent and controlled environment for evaluating the LAM's performance.

## 7.3 Baselines

To benchmark the performance of LAM, we compared it against two baseline models: GPT-4o and GPT-4o Mini. These models are widely recognized for their robust natural language processing and reasoning capabilities, making them popular choices in the development of GUI agents. To ensure consistency in evaluation, the `top_p` and `temperature` hyperparameters were set to 0 for both baseline models.

To further examine the impact of input modalities, we conducted an ablation study comparing performance with and without the inclusion of screenshots. Notably, LAM processes only textual inputs, excluding screenshots, while the baseline models were evaluated using both textual and visual modalities.

---

[6]https://azure.microsoft.com/en-us/products/virtual-machines/dedicated-host
[7]https://learn.microsoft.com/en-us/windows-server/administration/windows-commands/mstsc

### 7.4 Evaluation Metrics

We employ the following metrics to comprehensively evaluate the performance of LAM: **Task Success Rate (TSR):** The percentage of tasks successfully completed out of the total tasks attempted. Task success is determined by an evaluation agent using GPT-4o, which assesses the full task completion trajectory, including plans, action sequences, and screenshots, to verify task completion. **Task Completion Time:** The total time taken to complete each task, measured from the initial request to the final action. **Task Completion Steps:** The total number of action steps performed by the agent to successfully complete each task. **Average Step Latency:** The average time taken per action step, reflecting the model's efficiency in generating and executing each action.

To ensure the reliability of our LLM-based automatic evaluation, we conducted a small-scale human study focused on validating its accuracy and consistency. Specifically, we randomly sampled 100 task completions from the test set, spanning a diverse range of task types and difficulty levels. Each sample was independently reviewed by three human annotators with domain familiarity, who were asked to judge task success following the same criteria used in our automated evaluation. The results demonstrated a 91% agreement between the majority human judgment and the LLM-based labels. Furthermore, the inter-annotator agreement, measured by Cohen's kappa, was 0.85, indicating near-perfect consistency among human raters and strengthening the case that the automated evaluation aligns well with human judgment. We ensured cross-validation among annotators, and disagreements were resolved through consensus discussions. These findings confirm that our LLM-based evaluation is a valid and scalable proxy for human judgment in this task setting. Quantitative results, annotator agreement, and evaluation setup for the human study are provided in Appendix G.

### 7.5 Experimental Analysis

The experimental results are presented in Table 3. LAM achieves a TSR of 71.0%, demonstrating competitive performance compared to the GPT-4o models. While GPT-4o with visual inputs attains the highest TSR of 76.5%, slightly outperforming LAM, its reliance on visual data introduces significant trade-offs in efficiency. Notably, when visual inputs are excluded, GPT-4o's TSR drops to 63.0%, an 8.0 percentage point decrease compared to LAM. Similarly, GPT-4o Mini exhibits lower TSRs for both visual and non-visual settings (66.7% and 57.8%, respectively). These results underscore LAM's capability as a text-only model to maintain high task success rates, outperforming the text-only variants of the baseline models.

Efficiency is assessed through Task Completion Time and Average Step Latency, where LAM demonstrates clear superiority. LAM achieves the shortest Task Completion Time of **30.42 seconds**, substantially outperforming all baseline models. In comparison, GPT-4o without visual inputs records a completion time of 86.42 seconds, more than 2.84 times longer than LAM. GPT-4o with visual inputs fares even worse, with a completion time of 96.48 seconds. Although GPT-4o Mini models show slightly better efficiency than their larger counterparts, they remain less efficient than LAM, with completion times of 35.24 seconds (without visual inputs) and 46.21 seconds (with visual inputs).

LAM also excels in Average Step Latency, achieving the shortest time per action step at **5.41 seconds**. Without visual inputs, GPT-4o reduces its step latency to 12.84 seconds but still remains more than twice as slow as LAM. In comparison, GPT-4o with visual inputs exhibits the highest step latency at 19.36 seconds per step, more than triple LAM's latency. GPT-4o Mini models show moderate improvements but still fall short, with step latencies of 7.29 seconds (with visual inputs) and 5.88 seconds (without visual inputs).

These findings highlight LAM's strengths as a text-only model, offering a compelling balance of competitive accuracy and superior efficiency. It achieves rapid task completion and low latency without sacrificing performance, making it an effective solution for real-world applications. Its specialized training enables precise action inference and execution, underscoring the potential of LAMs to enhance automation and productivity in agent-based systems.

# 8 Limitations and Future Directions

Despite their promise, Large Action Models (LAMs) remain in an early stage, facing several key challenges that limit their deployment in real-world applications. We highlight three core areas requiring further research.

**Safety Risks** LAMs operate in both digital and physical environments, which introduces safety concerns not present in traditional LLMs. Erroneous actions—whether controlling software or physical systems—can result in real-world harm (Liu et al., 2024; Zhou et al., 2024). This necessitates robust safety mechanisms, such as action validation, formal verification, and rollback strategies (Zhang et al., 2023; Koo & Toueg, 1986). Future work should focus on fail-safe architectures that vet actions before execution.

**Ethical and Regulatory Concerns** As LAMs gain autonomy, questions of accountability, transparency, and bias become critical (Biswas & Talukdar, 2023; Li et al., 2023; Ferrara, 2024). Misinterpretation of user intent or inherited biases from training data can result in unfair or unsafe behavior. Regulatory compliance in sensitive domains (e.g., healthcare, finance) further complicates deployment (Karabacak & Margetis, 2023). Future directions include developing interpretable decision frameworks and establishing ethical guidelines and compliance standards for safe LAM deployment.

**Scalability and Generalization** Current LAMs are highly specialized, often limited to narrow environments. Changes in application interfaces or system updates can break performance (Grosse et al., 2023; Zhang et al., 2024). Moreover, domain-specific data collection is costly and labor-intensive (Muennighoff et al., 2023). Future research should pursue transfer learning, few-shot adaptation, and automated data collection to enhance generalizability and reduce dependence on manual annotation.

## 8.1 Mitigation Strategies for Safe Deployment

To ensure the safe and ethical deployment of Large Action Models (LAMs), we adopt the following mitigation strategies: (1) **Environment Sandboxing:** Run LAMs in sandboxed environments to restrict access to critical system functions, minimizing the risk of unintended actions. (2) **Human-in-the-Loop Verification:** Require human approval for high-impact or irreversible actions to ensure oversight in sensitive contexts. (3) **Action Validation:** Validate predicted actions in real-time against environmental constraints or through simulation to prevent execution errors. (4) **Fail-Safe Defaults and Timeouts:** Use safe defaults (e.g., cancel or no-op) and timeout mechanisms to handle ambiguous or stalled executions. (5) **Progressive Deployment:** Start with low-risk or shadow deployments and gradually scale to full autonomy based on observed reliability.

These strategies follow best practices for trustworthy AI and are essential for deploying LAMs safely in dynamic, real-world environments.

# 9 Related work

## 9.1 Data of LAMs

Mind2Web (Deng et al., 2023) is the first dataset developed for web agents that follow natural language instructions to complete complex tasks across diverse websites. It includes task descriptions, action sequences, and webpage snapshots, offering rich data for training and testing models in various web-based scenarios. Rawles *et al.*, introduced a large dataset called Android in the Wild (AITW) (Rawles et al., 2023), which is designed specifically for training models to control Android devices. SeeClick (Cheng et al., 2024) combines web, mobile, and general GUI tasks, creating a dataset of over 1 million samples for training LAMs. Similarly, GUICourse (Chen et al., 2024b) and OmniACT (Kapoor et al., 2024) provide datasets across web, smartphone, and desktop platforms, containing detailed user requests, environmental states, and action sequences. These datasets are invaluable resources for training LAMs in specific domains and evaluating their task execution abilities.

Several benchmarks have also been developed to evaluate the capabilities of LAMs and their associated agents in different environments. WebCanvas provides 542 tasks with dynamic environments, designed to assess the task completion ability of web agents. AndroidWorld (Rawles et al., 2025) offers a fully functional Android environment, featuring 116 programmatic tasks across 20 real-world Android apps with reward signals for performance evaluation. WindowsArena (Bonatti et al., 2024) focuses on benchmarking LAMs within the Windows GUI, while OSWorld (Xie et al., 2024) extends this to a more diverse environment, encompassing Windows, macOS, and Ubuntu. These benchmarks provide standardized settings to measure and compare the effectiveness of LAMs and their agents in various real-world environments, enabling a unified evaluation framework for agentic models.

## 9.2 Training LAMs

Using both open and private domain-specific datasets, significant research efforts have been directed toward training LAMs for specialized purposes, enhancing the action inference abilities of traditional LLMs to enable automation and tangible real-world impact. For example, SeeClick (Cheng et al., 2024) and GUICourse (Chen et al., 2024b), in addition to releasing their own datasets, leverage these resources to train LAMs, grounding real-world data into models that effectively interact with their environments.

Hong *et al.*, trained an 18-billion-parameter visual language LAM, named CogAgent (Hong et al., 2024), which specializes in GUI understanding and navigation tasks across both PC and Android interfaces. By utilizing datasets like Mind2Web and AITW, CogAgent has been optimized for complex navigation and action execution tasks in diverse GUI environments. ScreenAI (Baechler et al., 2024) introduced a textual representation for user interfaces (UIs) to teach models how to understand and interact with UIs. This approach also facilitates automatic generation of large-scale training data, which is then used to pretrain and fine-tune models for a wide spectrum of tasks, including UI and infographic understanding and navigation. Additionally, Zhang *et al.*, released a series of large action models (xLAM) tailored for AI agent tasks (Zhang et al., 2025c), including five models with both dense and mixture-of-expert architectures. By unifying datasets from diverse environments, xLAM ensures consistency in data format, simplifying model training and enhancing generalization across multiple benchmarks. These models have achieved outstanding performance in diverse scenarios, demonstrating the capability of LAMs to extend beyond traditional LLMs and perform complex real-world tasks.

## 9.3 Agents with LAMs

With the development of LAMs, researchers have integrated these models into real-world agent systems, which provide the necessary components and workflows to ensure effective interaction between LAMs and their environments, enabling them to fulfill user requests efficiently. As a pioneer, Zhang *et al.*, demonstrated that GPT-V can serve as a capable LAM for web navigation when coupled with appropriate agent techniques and tools, revealing the potential of LAMs in complex web interactions. In the mobile domain, MobileAgent (Wang et al., 2024a) and AppAgent (Zhang et al., 2025b) focus on automating tasks within Android applications by leveraging GUI agents. These systems demonstrate how LAMs can power task automation on mobile platforms, transforming how users interact with applications.

One of the most advanced systems, UFO (Zhang et al., 2025a), is a UI-focused agent designed for automating tasks on the Windows OS, further enhanced with APIs (Lu et al., 2024). UFO is composed of two key components: a HostAgent that decomposes user requests into subtasks and an AppAgent that executes these subtasks within individual applications. This architecture significantly enhances UFO's capability to handle cross-application tasks seamlessly, providing robust task automation across diverse software environments. In parallel, ScreenAgent (Niu et al., 2024), Cradle (Tan et al., 2024), OS-Copilot (Wu et al., 2024), and MMAC-Copilot (Song et al., 2024) also focus on automating UI tasks in desktop environments. Notably, Cradle and OS-Copilot push the boundaries by enabling agents to learn from their experiences and self-evolve over time, further enhancing their effectiveness and autonomy.

## 10 Conclusion

"Actions speak louder than words." The transition from generating language responses to executing tangible actions marks the evolution of large language models into large action models, enabling them to make real-world impacts, a critical step towards achieving AGI. This technical report provides a comprehensive introduction to LAMs, covering their conceptual foundations, system architecture, and the step-by-step process of developing a LAM—from data collection to model training and deployment in real-world agent systems. We use the Windows OS environment and its GUI agent UFO, as a case study to demonstrate how to build a LAM from the ground up. Detailed implementation strategies and evaluation results are presented to offer practical insights into this process.

However, despite progress, the development of high-quality LAMs is still in its early stages, with several limitations remaining. These include the extensive need for training data and computational resources, inference latency, and the risk of errors during real-world execution. While current LAMs have shown potential, there is substantial room for improvement. We anticipate that as these challenges are addressed, more sophisticated and reliable LAM applications will emerge, bringing us closer to fully autonomous systems capable of meaningful action in complex environments.

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

## A    Template Word files

Figure 10,  11, and  12 show three template word file examples used in the instantiation phase when converting task-plan data to task-action data.

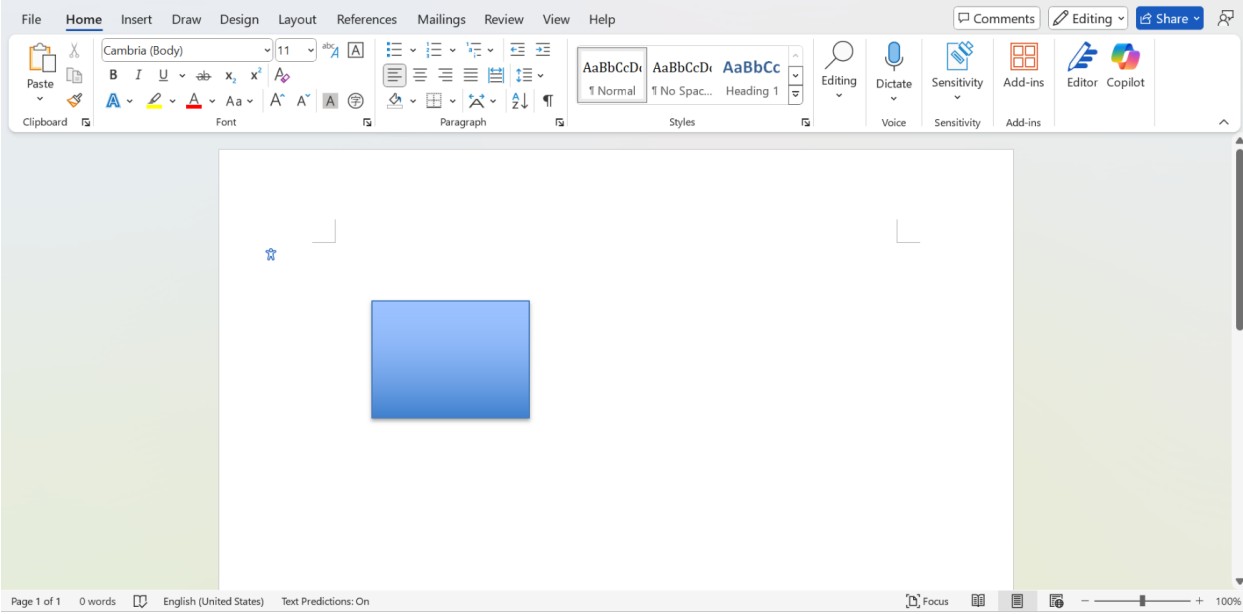

Figure 10: A word template file with the description "A doc with a rectangle shape."

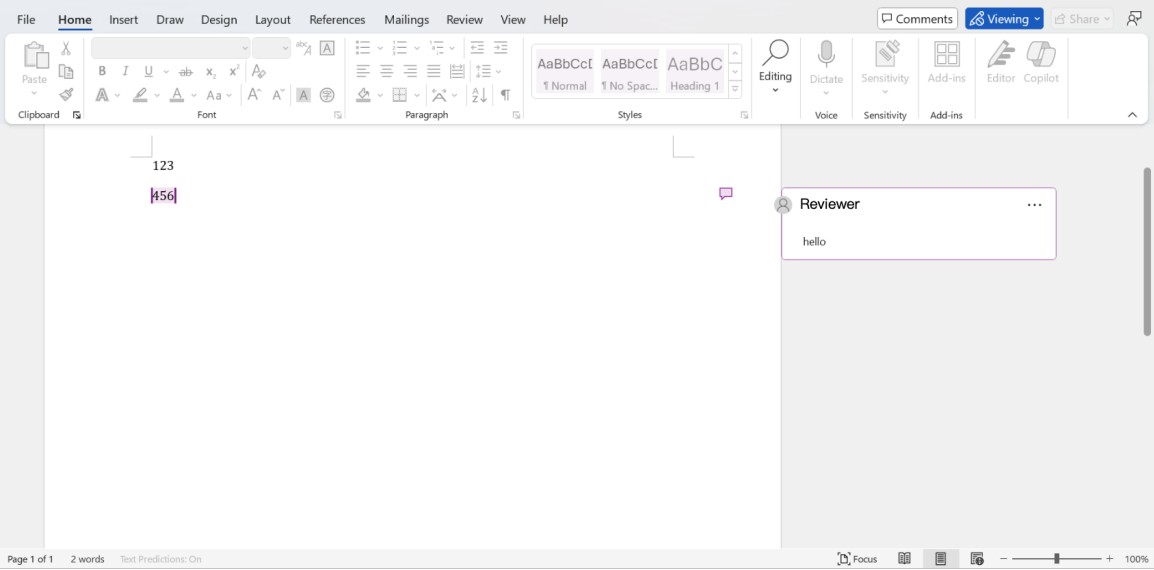

Figure 11: A word template file with the description "A doc with comments and reviewer."

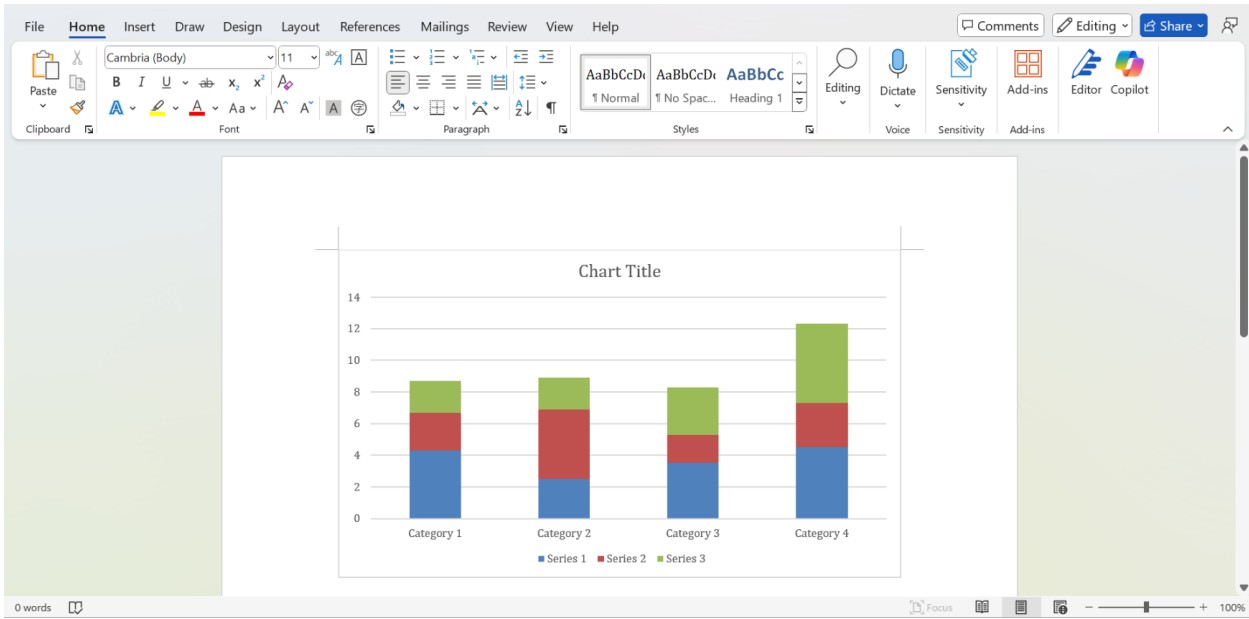

Figure 12: A word template file with the description "A doc with a chart."

# B Prompts

## B.1 Instantiation

The instantiation prompt used in the instantiation phase when converting task-plan data to task-action data.

```
system: |-
  You are a Agent Task Creator and planer.
  You will receive a <Given Task> that is abstract and your objective is to
      instantiate this task, and give the step-by-step actions to take.
  - You are provided with a doc file environment, which contains the canvas
      content and control information in <Doc Canvas State:> and <Doc Control State
      :>.
  - You should review the doc canvas content and control information to detail the
       <Given Task> to a <New Task>.The control information is in a dict tree of
      available control items format.
  - You are provided with <Available Actions>, you should review the acions
      carefully and choose the most suitable ones step-by-step <Action Plan>.
  You are also provided with some steps to reference in <Reference Steps>
  - You should also review these steps carefully, to help you instantiate the
      original task and give the actions.

  ## Control item
  - The control item is the element on the page that you can interact with, we
      limit the actionable control item to the following:
  - "Button" is the control item that you can click.
  - "Edit" is the control item that you can click and input text.
  - "TabItem" is the control item that you can click and switch to another page.
  - "ListItem" is the control item that you can click and select.
  - "MenuItem" is the control item that you can click and select.
  - "ScrollBar" is the control item that you can scroll.
  - "TreeItem" is the control item that you can click and select.
```

- `"Document"` is the control item that you can click and select text.
- `"Hyperlink"` is the control item that you can click and open a link.
- `"ComboBox"` is the control item that you can click and input text. The Google
  search box is an example of ComboBox.

## Available Actions on the control item
- All the available actions are listed below:
{apis}

## The requirements for <New Task>
1. The <New Task> must based on the given task.
2. The <New Task> must be able to be completed step-by-step by a Windows
   Operating System or an Application on Windows platform.
3. You should try your best not to make the <New Task> become verbose, <New Task
   > can only add up to 50 words into #Given Task#.
4. The detailed target in <New Task> should be specific and clear based on the
   doc canvas content and control information.
5. The <New Task> should be able to implemented by the available controls and
   actions.

## The requirements for <Action Plan>
1. The <Action Plan> should be step-by-step actions to take in the doc file
   environment.
2. Each action should be in the available actions from <Available Actions>.
3. Each action should be generated with a `"step"` description which is the
   function description of the action.

## Response Format
- You are required to response in a JSON format, consisting of several distinct
  parts with the following keys and corresponding content:
  {{
    `"observation"`: <Outline the observation of the provided doc file environment
        based on the given Canvas State and Control State>,
    `"thought"`: <Outline your thinking and logic of your New Task and the actions
        to take,consider the observation of environment and avaiable controls
        actions>,
    `"new_task"`:<Give the detailed New Task based on Given Task and the
        observation of doc environment>,
    `"actions_plan"`:<Give the detailed step-by-step actions plan based on the
        Available Actions and the observation of doc environment.,
    The format should be a list of action call format separated by `"\n"`>
  }}

### Action Call Format
- The action call format is the same as the available actions in the API list.
   You are required to provide the action call format in a JSON format:
  {{
  `"step"`: <The step description the function of the action,which is also the
      subtask completed by the current action>
  `"controlLabel"`: <Specify the precise annotated label of the control item to be
       selected, adhering strictly to the provided options in the field of `"label`
      `"` in the <Doc Control State:>. If you believe none of the control item is
      suitable for the task or the task is complete, kindly output a empty string
       .>
  `"controlText"`: <Specify the precise control_text of the control item to be
      selected, adhering strictly to the provided options in the field of `"
      control_text"` in the <Doc Control State:>.The control text must match
      exactly with the selected control label. If the function to call do not

```
       need specify controlText or the task is complete ,you can kindly output an
        empty string .
     If the function to call need to specify controlText and none of the control
        item is suitable for the task ,you should input a possible control name.>
    "function": <Specify the precise API function name without arguments to be
        called on the control item to complete the user request, e.g.,
        click_input. Leave it a empty string "" if you believe none of the API
        function is suitable for the task or the task is complete.>
    "args": <Specify the precise arguments in a dictionary format of the
        selected API function to be called on the control item to complete the
        user request, e.g., {{"control_id":"1","button": "left", "double": false
        }}. Leave it a empty dictionary {{}} if you the API does not require
        arguments, or you believe none of the API function is suitable for the
        task, or the task is complete.>
  }}

  e.g.
    {{
        "step": "change the borders",
        "controlLabel": "",
        "controlText": "Borders",
        "function": "click_input",
        "args": {{
            "button": "left",
            "double": false
        }}
    }}

    {{
      "step": "change the borders",
        "controlLabel": "101",
        "controlText": "Borders",
        "function": "click_input",
        "args": {{
            "control_id": "101",
            "button": "left",
            "double": false
        }}
    }}

    {{
        "step": "select the target text",
        "controlLabel": "",
        "controlText": "",
        "function": "select_text",
        "args": {{
            "text": "Test For Fun"
        }}
    }}

- The <actions_plan> field must be strictly in a format separated each action
   call by "\n". The list format should be like this: "action call 1\naction
   call 2\naction call 3"
- If you think the original task do not need be detailed , you can directly
   copy the original task to the "new_task".
- You should review the apis function carefully and if the function to call need
    to specify target control ,the "controlText" field
cannot be set empty.
```

```
    - The "step" description should be consistent with the action and also the
        thought.

    ## Here are some examples for you to complete the user request:
    {examples}

    ## Tips
    - Read the above instruction carefully. Make sure the response and action
        strictly following these instruction and meet the user request.
    - Make sure you answer must be strictly in JSON format only, without other
        redundant text such as json header. Your output must be able to be able to be
         parsed by json.loads(). Otherwise, it will crash the system and destroy the
        computer.
    - Your task is very important to improve the agent performance. I will tip you
        200$ if you do well. Thank you for your hard work!

user: |-
  <Given Task:> {given_task}
  <Reference Steps:> {reference_steps}
  <Doc Canvas State:> {doc_canvas_state}
  <Doc Control State:> {doc_control_state}
  <Your response:>
```

## B.2 Evaluation

The instantiation prompt used in the evaluation phase when converting task-plan data to task-action data.

```
system: |-
  You are an evaluator who can evaluate whether an agent has successfully
      completed a task in the <Original Request>.
  The agent is an AI model that can interact with the desktop application and take
       actions.
  The thought of agent plan is provided in the <Thought>.
  You will be provided with a task and the <Execution Trajectory> of the agent,
      including the agent actions that have been taken, and the change of
      environment.
  You will also be provided with a final canvas state in <Final Env Status>.
  You will also be provided with a canvas difference in <Canvas Diff>.
  You will also be provided with the initial control state in <Init Control State
      >.
  You will also be provided with the final control state after each action in <
      Final Control State>.

  Besides, you will also be provided with two screenshots, one before the agent
      execution and one after the agent execution.

  Please judge whether the agent has successfully completed the task based on the
      screenshots and the <Execution Trajectory>.You are required to judge whether
      the agent has finished the task or not by observing the screenshot
      differences and the intermediate steps of the agent.

  ## Execution trajectory information
  Here are the detailed information about a piece of agent execution trajectory
      item:
  - number: The number of action in the execution trajectory.
  - action: The action that the agent takes in the current step. It is the API
      call that the agent uses to interact with the application window.
```

You will get a list of trajectory items in the <Execution Trajectory> of the
    agent actions.

### Control State

- A control item is the element on the page that you can interact with, we limit
    the actionable control item to the following:
- "Button" is the control item that you can click.
- "Edit" is the control item that you can click and input text.
- "TabItem" is the control item that you can click and switch to another page.
- "ListItem" is the control item that you can click and select.
- "MenuItem" is the control item that you can click and select.
- "ScrollBar" is the control item that you can scroll.
- "TreeItem" is the control item that you can click and select.
- "Document" is the control item that you can click and select text.
- "Hyperlink" is the control item that you can click and open a link.
- "ComboBox" is the control item that you can click and input text. The Google
    search box is an example of ComboBox.
- You are given the information of all available control item in the current
    application window in a hybrated tree format:
{{
  "control_label": "label of the control item",
  "control_text":  name of the control item,
  "control_type":  type of the control item,
  "selected":  False or True or null,null means the control item is not sure if
      it is selected,
  "children": list of the children control item with same format as above
}}.

### Canvas State Format
The canvas state is in the xml format which is transformed from the document
    object model (DOM) of the canvas area.
The canvas diff is the difference of the canvas area before and after the action
    , which is in the format of the difference of the xml of the canvas area.
Here is an example of xml of a canvas,which show the text content in document:

{{"w:document":{{"@mc:Ignorable":"w14w15w16sew16cidw16w16cexw16sdtdhw16duwp14","
    w:body":{{"w:p":{{"w:pPr":{{"w:rPr":{{"w:rFonts":{{"@w:hint":"eastAsia"}}},"w:
    color":{{"@w:val":"92D050"}},"w:kern":{{"@w:val":"2"}},"w:sz":{{"@w:val":"24"
    }},"w:szCs":{{"@w:val":"24"}},"w:lang":{{"@w:val":"en-US","@w:eastAsia":"zh-
    CN","@w:bidi":"ar-SA"}},"w14:ligatures":{{"@w14:val":"standardContextual"
    }}}},"w:spacing":{{"@w:after":"160","@w:line":"278","@w:lineRule":"auto"}},"w
    :color":"000000"}},"w:r":{{"w:rPr":{{"w:rFonts":{{"@w:hint":"eastAsia"}},"w:
    color":{{"@w:val":"92D050"}},"w:highlight":{{"@w:val":"yellow"}},"w:kern":{{"
    @w:val":"2"}},"w:sz":{{"@w:val":"24"}},"w:szCs":{{"@w:val":"24"}},"w:lang":{{
    "@w:val":"en-US","@w:eastAsia":"zh-CN","@w:bidi":"ar-SA"}},"w14:ligatures":{{
    "@w14:val":"standardContextual"}}}},"w:t":"Hello"}}}},"w:sectPr":{{"w:pgSz"
    :{{"@w:w":"12240","@w:h":"15840"}},"w:pgMar":{{"@w:top":"1440","@w:right":"
    1440","@w:bottom":"1440","@w:left":"1440","@w:header":"720","@w:footer":"720"
    ,"@w:gutter":"0"}},"w:cols":{{"@w:space":"720"}},"w:docGrid":{{"@w:linePitch"
    :"360"}}}}}}}}}}

### Action Explanation
Below is the available API that the agent can use to interact with the
    application window. You can refer to the API usage to understand the agent
    actions.
{apis}

## Evaluation Items

You have 2 main items to evaluate:

1. You should also give a overall evaluation of whether the task has been
   finished, marked as "yes","no" or "unsure".
2. You should also give a overall evaluation of the quality of task,marked as "
   ambiguous","over-detailed" or "good".

Criteria for evaluation of the task completion:
1. The <Final Control State:> and <Final Env Status:> should be consistent with
   the task requirements.If the
controls or canvas content expected to be changed are not changed, the task is
   not completed.
2. The <Execution Trajectory> should be consistent with the task requirements.
   If the agent actions are not consistent with the task requirements, the task
   is not completed.
3. If any action in the <Execution Trajectory> is empty, the task is not
   completed.

Criteria for evaluation of the task quality:
1. The description of the <Original Request:> should be clear and unambiguous,
   without the meaning of "selection".
2. The description of the <Original Request:> should not be too detailed like
   step-by-step actions.

## Response Format

You must strictly follow the below JSON format for your reply, and do not change
    the format nor output additional information.
{{
    "task_quality": The quality of the <Original Request:>, which is "ambiguous/
        over-detailed/good",
    "task_complete": The evaluation of the task completion, which is "yes/no/
        unsure",
    "complete_judgement": your judgment of whether the task has been finished,
        and the detailed reasons for your judgment based on the provided
        information,
    "quality_judgement": your judgment of the quality of the task, and the
        detailed reasons for your judgment based on the provided information
}}

Please take a deep breath and think step by step. Observe the information
   carefully and analyze the agent execution trajectory, do not miss any minor
   details.
Rethink your response before submitting it.
Your judgment is very important to improve the agent performance. I will tip you
    200$ if you provide a detailed, correct and high-quality evaluation. Thank
   you for your hard work!

user: |-
  <Original Request:> {request}
  <Thought:> {thought}
  <Execution Trajectory:> {trajectory}
  <Canvas Diff:> {canvas_diff}
  <Init Control State:> {init_control_state}
  <Final Control State:> {final_control_state}
  <Final Env Status:> {final_status}

```
    <Your response:>
```

## C   Templates of training format

The following presents a template of the training data format. The parts enclosed in "" are fields that need to be filled. The "apis" field corresponds to the function information in the respective app, while "control_item" contains the control information of the app under the current screenshot. The "user_request" field captures the user's current request, "step_history" records the agent's previous trajectory history, and "previous_plan" outlines the agent's planning for the task in the previous state.

```
system: |-
  - You are a virtual assistant that can help users to complete their current
      requests by interacting with the UI of Window OS.
  - You are provided a list of control items of the current application window for
      reference
  - You are provided your previous plan of action for reference to decide the next
      step,the previous plan is the list of plan for the future actions made
      before the current action.
  - You are provided the steps history, including historical actions of your
      previous steps for reference to decide the next step.
  - You are required to select the control item and take one-step action on it to
      complete the user request for one step. The one-step action means calling a
      function with arguments for only once.
  - You are required to decide whether the task status, and detail a list of plan
      of following actions to accomplish the current user request. Do not include
      any additional actions beyond the completion of the current task.

  ## Control item
  - The control item is the element on the page that you can interact with, we
      limit the actionable control item to the following:
  - "Button" is the control item that you can click.
  - "Edit" is the control item that you can click and input text.
  - "TabItem" is the control item that you can click and switch to another page.
  - "ListItem" is the control item that you can click and select.
  - "MenuItem" is the control item that you can click and select.
  - "ScrollBar" is the control item that you can scroll.
  - "TreeItem" is the control item that you can click and select.
  - "Document" is the control item that you can click and select text.
  - "Hyperlink" is the control item that you can click and open a link.
  - "ComboBox" is the control item that you can click and input text.

  ## Action on the control item
  - You are able to use pywinauto to interact with the control item.
  {apis}

  ## Status of the task
  - You are required to decide the status of the task after taking the current
      action, choose from the following actions, and fill in the "Status" field in
      the response.
    - "CONTINUE": means the task is not finished and need further action.
    - "FINISH": means the current task is finished for the AppAgent and no further
        actions are required.

  ## Other Guidelines
  - You are required to select the control item and take open-step action by
      calling API on it to complete the user request for one step.
```

```
      - You are required to response in a JSON format , consisting of 7 distinct parts
         with the following keys and corresponding content :
      {{
      "thought": <Outline your thinking and logic of current one -step action
         required to fulfill the given request . You are restricted to provide you
         thought for only one step action. >
      "control_label": <Specify the precise annotated label of the control item to
         be selected , adhering strictly to the provided options in the field of "
         label" in the control information . If you believe none of the control item
         is suitable for the task or the task is complete , kindly output a empty
         string .>
      "control_name": <Specify the precise control_text of the control item to be
         selected , adhering strictly to the provided options in the field of "
         control_text" in the control information . If you believe none of the
         control item is suitable for the task or the task is complete , kindly
         output a empty string . The control text must match exactly with the
         selected control label. >
      "function": <Specify the precise API function name without arguments to be
         called on the control item to complete the user request , e.g., click_input .
          Leave it a empty string "" if you believe none of the API function is
         suitable for the task or the task is complete. >
      "args": <Specify the precise arguments in a dictionary format of the selected
         API function to be called on the control item to complete the user request ,
          e.g., {{"button": "left", "double": false}}. Leave it a empty dictionary
         {{}} if you the API does not require arguments , or you believe none of the
         API function is suitable for the task , or the task is complete. >
      "status": <Specify the status of the task given the action. >
      "plan": <Specify the following list of plan of action to complete the user
         request . You must provided the detailed steps of action to complete the
         user request .If you believe the task is finished and no further actions are
          required after the current action , leave it an empty list. >
      }}
user: |-
   <Available Control Item:> {control_item}
   <User Request:> {user_request}
   <Previous Actions:> {step_history}
   <Previous Plans:> {previous_plan}

assistant: |-
   {output}
```

# D   Evaluation Prompt for Task-Plan

The evaluation prompt for results from LAM[1] after task-plan pretraining.

```
You are a helpful and precise assistant for checking the quality of the answer . We
    would like to invite you to evaluate the performance of two AI assistants in
   answering a users question in <Question >. These two answers are in <Answer1 >
   and <Answer2 >, respectively . Your evaluation will contain five sub - evaluation
   tasks:
1. Can <Answer1 > solve the users question ?
    - Your answer should be "Yes" or "No".
2. Can <Answer2 > solve the users question ?
    - Your answer should be "Yes" or "No".
3. Both two answers contain a list of steps marked by numbers . Your task is to
    extract action items from the provided steps in both answers . The action item
    is defined like a combination of action and element . Compare the action items
```

```
    to identify similarities. Output the similar action items. Count the count of
    similar action items.
    - Your answer should contain the extracted two action item sets (in the format
        as a list of string).
    - Your answer should contain the set of similar action items (in the format as
        a list of string). Similar action items are those sharing similar intent
        or achieving similar goals. Each similar action pair in the list should be
        in the format of "similar action item from action item set1 / similar
        action item from action item set2"
    - Your answer should contain the count of similar action items.
4. Which assistant provides a more helpful response?
    - Your answer should be "1" or "2", where "1" represents <Answer1> and "2"
        represents <Answer2>.
    - Your answer should contain the reason(s) for your choice. You should not
        focus on the length of the answer or the details of the answer, but you
        should focus on whether the steps could solve the users question and the
        quality of the steps.

Your output should be in the following format in json:
{{
    "Subtask1": "Yes" or "No",
    "Subtask2": "Yes" or "No",
    "Subtask3": {{
        "Action items in Answer1": ["action item 1", "action item 2", ...],
        "Action items in Answer2": ["action item 1", "action item 2", ...],
        "Similar action items": ["similar action item 1", "similar action item 2",
            ...],
        "Count of similar action items": 2
    }},
    "Subtask4": {{
        "More helpful assistant": "1" or "2",
        "Reason": "reason for your choice"
    }}
}}

Here is the users question <Question>: {question}
The first answer <Answer1> is: {answer1}
The second answer <Answer2> is: {answer2}
```

## E   LAM Training Objectives

The problem is formally structured into two key objectives: *(i)* task-plan pretraining and *(ii)* decision-making training.

*Task-plan pretraining* aims to enable the LAM to map a given task description to a structured sequence of plans necessary for accomplishing the task. The primary objective of this component is to generate accurate and coherent plans. The training dataset consists of task-plan pairs, defined as:

$$\mathcal{D}_{\text{plan}} = \{(t_i, P_i)\}_{i=1}^{N}$$

where $t_i$: The task description, $P_i$: A sequence of plans to complete the task.

In *decision-making training*, the dataset consists of task-action trajectories, defined as::

$$\tau = \{(s_1, a_1), (s_2, a_2), \ldots, (s_T, a_T)\}$$

where:

- $s_t$ (state at time step $t$), comprising:

- **Task description**: A high-level summary of the task.
- **Step ID**: The current step in the task sequence.
- **Observations**: Information including control elements and the current canvas state.
- **Thoughts**: Model-generated reasoning for the current step.
- **Previous actions and plans**: The sequence of actions and plans from prior steps.

- $a_t$ (action taken at time step $t$), consisting of:

  - **Thought**: Model's reasoning for the action.
  - **Control label**: A label for the control element.
  - **Control name**: The name of the control to interact with.
  - **Function name**: The specific function invoked by the action.
  - **Arguments**: Parameters passed to the function.
  - **Status**: Indicates action's progress, either ongoing (*Continue*) or completed (*Finish*).

The objective of decision-making training is to train the LAM to predict the appropriate action $a_t$ for a given state $s_t$ at each time step. This enables the model to map input states to corresponding actions across the sequence of steps required to accomplish the task.

**Broader Impact Statement**

The Large Action Model (LAM) framework provides a novel approach for training agents in scenarios where no labeled data exists. By combining task-plan pretraining, imitation learning, and autonomous self-boosting with reward-based fine-tuning, LAM enables agents to progress from zero prior knowledge to proficient task execution. This capability is particularly valuable in data-scarce domains, offering a scalable method to bootstrap agent learning using structured knowledge from resources like help documentation. Such an approach makes it feasible to deploy AI systems in specialized environments with limited access to traditional training datasets.

The potential impact of LAM extends to a wide range of applications, including automation of complex user interactions, optimization of industrial workflows, and enhancement of accessibility technologies. However, the autonomous learning capabilities of LAM also raise ethical concerns. Misaligned or incorrect actions in critical applications could have unintended consequences, and questions of accountability must be carefully addressed.

To mitigate risks, it is crucial to implement robust evaluation mechanisms and safeguards during deployment. Transparency, alignment with user intent, and clear oversight are essential to ensure responsible use. By addressing these challenges, the LAM framework offers a pathway to extend the reach of AI into domains previously considered inaccessible due to data limitations.

# F   Data Deduplication and Train-Test Similarity Analysis

## F.1   Semantic Deduplication and Filtering Pipeline

To improve training efficiency and ensure data diversity, we applied a semantic deduplication pipeline inspired by Self-Instruct. Specifically: We first parsed all raw instruction data and removed irrelevant samples (e.g., non-English instructions, malformed data, smartphone-related tasks). We began with 76,851 raw tasks samples. After applying heuristic filters and semantic deduplication (using Sentence-BERT all-MiniLM-L6-v2 (Reimers & Gurevych, 2021) with a cosine similarity threshold of 0.95), we removed 179 samples, resulting in 76,672 cleaned tasks.The final split used for training and evaluation is 1,757 training tasks and 435 test tasks.

**ROUGE-L Distribution Between Train and Test Instructions**

We computed the ROUGE-L score between each test instruction and its closest training instruction. The summary statistics are Mean ROUGE-L: 0.27, 25th percentile: 0.19, 75th percentile: 0.34, Standard deviation: 0.13.

### F.2 Train-Test Instruction Pairs with ROUGE-L Scores

We include several representative train-test pairs below to show their semantic difference.

**PAIR 1: ROUGE-L Score = 0.2000   Test:** *"Attach a digital signature in Word by navigating to the 'Insert' tab, selecting the 'Signature Line' option, and following the prompts to insert and sign the signature line."*
**Train:** *"Select the 'text to edit' and attempt to access font settings to scale the text within the limitations of the available controls."*

**PAIR 2: ROUGE-L Score = 0.2069   Test:** *"Access the 'Add-ins' feature within Microsoft Word to explore or manage add-ins."*
**Train:** *"Switch to 'Outline' view in Microsoft Word to make structural changes, such as rearranging paragraphs."*

**PAIR 3: ROUGE-L Score = 0.1818   Test:** *"Edit the opened document in Word, assuming it is the Google Doc that has been downloaded and opened."*
**Train:** *"Use the 'Search' control to initiate a search for a 'chore checklist' template in Word."*

**PAIR 4: ROUGE-L Score = 0.2759   Test:** *"Navigate to the 'Insert' tab in Word as the initial step towards inserting two images side by side."*
**Train:** *"Locate and access the 'Mailings' tab in the Word document interface."*

**PAIR 5: ROUGE-L Score = 0.4000   Test:** *"Click on the 'Insert' tab in Word to begin the process of auto-inserting a paragraph using a command."*
**Train:** *"Navigate to the 'Insert' tab in Word as the initial step to use the 'entspricht' symbol."*

**PAIR 6: ROUGE-L Score = 0.1250   Test:** *"Save a Word document and open the font dialog box for more options."*
**Train:** *"Navigate to the 'Layout' tab in Word to begin the process of removing page numbers from a specific section."*

These examples confirm that although some instructions share interface elements (e.g., "Insert" tab), they are semantically and functionally distinct, thus preserving the integrity of our train-test split. The distribution of ROUGE-L score between train and test is in Figure 13.

## G   Human Evaluation Protocol and Results

To validate the reliability of our LLM-based automatic evaluation, we conducted a human study involving three trained annotators. Each annotator independently reviewed a set of 100 randomly sampled task completions. For each sample, annotators answered the following question:

> *Does the action sequence complete the intended task as described, given the initial application state?*

They provided a rating of `Yes` and `No` for each task. Each task was evaluated based on the task description, initial state (screenshots and XML), and the action sequence executed by the agent.

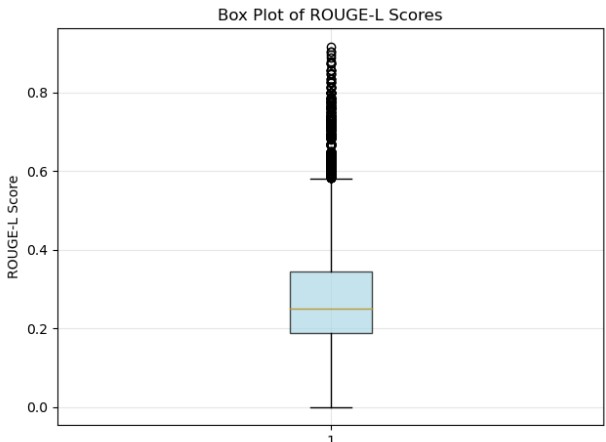

Figure 13: Distribution of ROUGE-L score between train and test.

**Quantitative Summary:**

- **Mean agreement with LLM-based evaluation:** $91\%$

- **Inter-annotator agreement (Cohen's $\kappa$):** $0.85$

- **Rating distribution:**
    - `Yes:` $83.3\% \pm 4.7\%$
    - `No:` $16.7\% \pm 4.7\%$

These results demonstrate strong alignment between human judgment and LLM-based scoring. The high Cohen's $\kappa$ score (0.85) indicates near-perfect inter-annotator reliability.

