# OpenReview forum: "Large Action Models: From Inception to Implementation"
_TMLR — Accepted by TMLR_

### Review · Reviewer_CeVy · 2025-02-07

**Summary Of Contributions:**

This paper introduces an approach for fine-tuning Large Action Models (LAMs) from Large Language Models (LLMs), specifically focusing on models that can execute actions in real-world environments. The methodology consists of several key steps. First, the authors curate a dataset by scraping task descriptions from multiple sources, including WikiHow and Bing search queries. They then leverage LLMs to generate detailed plans and step-by-step action sequences for these tasks. To enhance the dataset's complexity and coverage, they use LLMs to create additional variations with added constraints, expanding the dataset by 50% to reach a total of 76,672 tasks (29,182 original + 47,490 augmented). The authors ensure data quality through validation: a subset of tasks are manually paired with an appropriate environment (such as templated Word documents) and verified for action executability. They also implement an LLM-based evaluation system to verify whether the generated plans and action sequences successfully accomplish their intended tasks. To demonstrate their approach, the authors present a case study using Microsoft Word as the target environment. The fine-tuning process involves four distinct phases: three stages of supervised fine-tuning focused on plans and action trajectories, followed by a final phase of offline reinforcement learning using a reward model. They choose Mistral 7B as their base model for fine-tuning. The authors evaluate the resulting LAM both offline and online. In offline evaluations (without environment interaction), their model achieves comparable performance to GPT-4o on general tasks and superior performance on Word-specific tasks, despite its significantly smaller size. In online testing, the LAM outperforms GPT-4o in text-only scenarios. However, when GPT-4o has access to image inputs, it achieves better performance than LAM, albeit at the cost of longer processing times – approximately three times slower than LAM. The authors note that their mini version operates at about 1.5 times the speed of the standard implementation.

**Audience:**

Yes

**Claims And Evidence:**

No

**Requested Changes:**

*Required changes for acceptance*:
- Discuss how you can guarantee that the test set is held-out for evaluation, and that it is sufficiently different from training data.
- A small manual evaluation to confirm the automatic evaluation is reasonable
- More discussion on the trade-offs of this whole pipeline compared to simply using GPT-4o out of the box. Even though GPT-4o is almost 3 times slower at inference time, it does not require the whole task-specific data collection and training pipeline. Further, when using images, it seems like GPT-4o is still better than the LAM in the case study.
- Shorten the paper significantly
- Make sure all claims have the required evidence

*Recommended changes, not required for acceptance*
- Discuss why step precision is a reasonable metric, e.g. when GPT4o gets a lower step precision but still a task success, are the steps wrong or just different also valid steps?
- Typo: *"where we first The reward model (RM) is built using LAM3"* section 4.4
- Would be useful to give a one-line summary of how ReAct works to generate succesful trajectories (Section 4.3)

Examples of claims without evidence:
- *"This evaluation step ensures that only valid, accurate task-action data is included in the training dataset, contributing to the reliability and robustness of the LAM."* -> the evaluation is done automatically with an LLM, so it does not ensure validity or accuracy.
- *".. gains a robust task-agnostic planning capability"* -> the model does not gain a task-agnostic planning capability as far as I can see, it's in task-conditioned throughout the paper
- Several times it is mentioned you develop a LAM from scratch, but then the only experiment you do is fine-tuned from a Mistral base model

**Strengths And Weaknesses:**

## Strengths

- The authors present quite some detail on their data collection and pre-processing method.
- The authors collect a large and high-quality dataset, and present an interesting method to ensure validity of the action trajectories
- The proposed method achieves a high performance on a Word interaction task, comparable to GPT-4o using a 7B model.

## Weaknesses

**Experimental setup**

The experimental is, as far as I can judge, not sound:
- It is unclear to me how the offline evaluation tasks are selected, and whether the LAM is trained on them or not.
- Although the 435 Word environment trajectories are held out from the training set, given that they are generated by LLMs or scraped from the internet, it is unclear how different they are from the training tasks.
- Except comparing to GPT-4o out of the box, no baselines are evaluated.
- Evaluations are all done automatically using LLMs, I would expect at least a handful of human evaluations to confirm that the LLM evaluator is a reasonable proxy for actual accuracy.
- The main improvement of LAM over GPT-4o is on step precision. It is not clear to me why step precision is a reasonable metric to evaluate on. One can presumably achieve task success using different trajectories. The result in Section 5.2.2 then in my opinion does not show that LLMs have limitations in executing actions but rather shows that you can distill GPT-4o performance into smaller action models. It seems more likely to me that the LAM achieves a higher step precision because it is trained on similar steps as it is tested on, whereas GPT-4o was not.
- A lot of discussion in the introduction about LLM's failure to do complex planning, but it seems like your method is only tested on very short-term planning (less than 2 actions on average per trajectory for Phase 2 training (top of page 15))

**Applicability of the method**.
In essence, you are doing a lot of domain-specific data collection and training to get a model to perform well on a single environment. This means that likely for each new environment you need to repeat most of your pipeline, whereas GPT-4o can seemingly achieve similar results out of the box. This makes the method less scalable.

**Writing**.
The paper is 25 pages before references/appendix, but it should be much shorter. There is a lot of repetition throughout the paper that can be removed or condensed. For example, the task-plan data collection in section 3, *"This data is used to fine-tune the model to generate effective plans and improve its high-level reasoning and planning capabilities"* and *"The task-plan data aims at enhancing the model’s high-level planning capabilities, allowing it to generate detailed, step-by-step plans based on user requests."* a few sentences later. I would say the first 8 pages can be condensed into at most 3 pages. There's other issues with the writing, some of the English requires some work (e.g. Figure 1), the citations are not formatted correctly, and sometimes the writing is too unscientific for an academic article (claims are made without evidence, some examples given below).

---

> ### Author Response · Authors · 2025-05-30
>
> We sincerely thank the reviewers for their thoughtful and constructive feedback.
>
> **1. Required changes for acceptance:**
>
> **a. Clarification on Evaluation and Data Overlap**
>
> We appreciate the reviewer’s concern regarding the separation between training and test data, as well as the generalization of the model to novel tasks. We took several rigorous steps to ensure that the evaluation set is both held-out and semantically distinct from the training set:
>   - Redundancy Removal Before Splitting: Before partitioning the dataset, we applied semantic deduplication to eliminate redundant or highly similar tasks. This step ensures that no rephrased or paraphrased version of a training task appears in the evaluation set.
>   - Random Sampling After Cleaning: We then randomly sampled the held-out test set from the remaining data. The test tasks were never seen during model training or used for model selection.
>   - Semantic Similarity Validation: To further ensure that the test set provides a robust measure of generalization, we followed the methodology of Self-Instruct (Wang et al., ACL 2023) [1] and computed the ROUGE-L similarity between the seed instructions in the training and test sets. We found the average ROUGE-L score to be 0.27, which is consistent with the similarity reported in that work and indicates that the test instructions are semantically distinct from the training distribution.
> We have revised the manuscript to include these methodological details to clearly demonstrate the rigor of our evaluation protocol and the generalization capacity of our model in **Section 5.1.1 with blue text**.
> [1] Wang, et al. "Self-Instruct" ACL 2023.
>
> **b. Human Evaluation to Validate LLM Judgments**
>
> To ensure the reliability of our LLM-based automatic evaluation, we conducted a small-scale human study focused on validating its accuracy and consistency. Specifically, we randomly sampled 100 task completions from the test set, spanning a diverse range of task types and difficulty levels. Each sample was independently reviewed by three human annotators with domain familiarity, who were asked to judge task success following the same criteria used in our automated evaluation.
> The results demonstrated a 91% agreement between the majority human judgment and the LLM-based labels. Furthermore, the inter-annotator agreement, measured by Cohen’s kappa, was 0.85, indicating near-perfect consistency among human raters and strengthening the case that the automated evaluation aligns well with human judgment.
> We ensured cross-validation among annotators, and disagreements were resolved through consensus discussions. These findings confirm that our LLM-based evaluation is a valid and scalable proxy for human judgment in this task setting.
> We have added this study and its quantitative results to **the revised version (Section 7.4 with blue text)**, along with example task completions and their corresponding human and LLM labels, to increase transparency and support the robustness of our evaluation pipeline.
>
> **c. Trade-offs Compared to GPT-4o**
>
> RC: Trade-offs Compared to Using GPT-4o Directly
> We thank the reviewer for highlighting this important trade-off. While GPT-4o offers strong zero-shot capabilities, LAMs are designed for real-world deployment where latency, cost, and task specialization matter.
> GPT-4o is approximately 3× slower at inference time in our benchmarks, latency and cost constraints in production systems often make this delay unacceptable,especially in high-throughput or interactive environments like GUI automation. In contrast, LAMs are specifically trained for the target domain using lightweight models (e.g., 7B scale), offering substantially lower latency and compute costs, enabling real-time deployment on local or edge systems.
> LAM, using a lightweight 7B model, achieves 71.0% task success, outperforming GPT-4o’s 63.0% in the same text-only setting, with significantly lower inference latency and cost—critical for high-frequency GUI automation.
> Beyond performance, LAMs enable continual self-improvement through interaction and reward-based fine-tuning, which frozen models like GPT-4o cannot achieve. As noted in DIGIRL [2], prompting-only agents are bounded by their pretrained distributions.
> While our current LAM is text-only, we plan to extend it to a Visual LAM. Notably, our text-only LAM already performs comparably to GPT-4o with image + text, showing the strength of domain adaptation.
> In summary, LAMs offer a more efficient, scalable, and adaptable solution for task-specific deployment, balancing performance and feasibility in production. We added this discussion **in Section 1 with blue text**.
> [2] Bai, Hao, et al. "DIGIRL" NeurIPS 2024.
>
> d: Shorten the paper significantly
>
> We have condensed the main content of the manuscript **from 25 pages to 20 pages** by removing redundant descriptions, streamlining the introduction and methodology sections, and focusing on core contributions.

---

> > ### Author Response · Authors · 2025-05-30
> >
> > **e. Make sure all claims have the required evidence**
> >
> > We appreciate the reviewer’s attention to ensuring that claims are well-supported and precise. We would like to clarify that the LLM-based evaluation is not used to determine task success in isolation or outside of the environment. Rather, it is applied only after a trajectory has been successfully executed in the real environment (e.g., Microsoft Word) to assess whether the final state matches the task objective, i.e., whether the user's intent has truly been fulfilled.
> > This evaluation acts as an automated proxy to validate whether an environment-grounded trajectory indeed completes the intended task, rather than judging model output abstractly.
> >
> > To support the reliability of this automated evaluation, we conducted a human study in which three annotators independently reviewed 100 executed task completions. The results show 97% agreement between human and LLM-based labels, with a Cohen’s κ of 0.85, indicating near-perfect consistency. We will clarify this setup and include these findings in the revised manuscript to ensure transparency and rigor in our evaluation methodology.
> >
> > Task-Agnostic Planning: We acknowledge that the model operates in a task-conditioned manner. The term “task-agnostic” has been removed or rephrased to more accurately reflect the model’s generalization across task types during pretraining, not at inference time.
> >
> > Training from Scratch: We clarify that while the LAM is fine-tuned from a Mistral-7B base model, the full action modeling pipeline (including data collection, grounding, and evaluation) is developed from scratch for this specific setting. We no longer claim the model is trained “from scratch” in the absolute sense.
> >
> >
> > **2. Recommended changes, not required for acceptance**
> >
> > **a. Inclusion of More Baselines**
> >
> > We appreciate the reviewer’s suggestion and agree that including additional baselines strengthens the evaluation. In the revised manuscript, we have added results for two recent and competitive models: DeepSeek R1 (67.1B) and OpenAI o1-mini, as shown in Table 8 (Offline Evaluation).
> > ### Table: Offline performance comparison across different models and metrics on decision making
> >
> > | **Metric**                    | **LAM¹** | **LAM²** | **LAM³** | **LAM⁴** | **GPT-4o** | **GPT-4o Mini** | **DeepSeek R1** | **O1-mini** |
> > |------------------------------|----------|----------|----------|----------|------------|------------------|------------------|-------------|
> > | **Object Acc (%)**           | 39.4     | 85.6     | 87.4     | **87.8** | 73.2       | 68.4             | 63.4             | 71.8        |
> > | **Operation Acc (%)**        | 59.9     | 97.3     | 97.7     | **97.7** | 94.2       | 89.6             | 87.8             | 88.5        |
> > | **Status Acc (%)**           | 32.7     | 97.8     | 98.2     | **99.0** |      65.3| 60.2               | 68.2             | 59.4        |
> > | **Step Success Rate (SSR) %**| 33.0     | 83.6     | 85.9     | **86.2** | 68.8       | 62.7             | 59.4             | 69.3        |
> > | **Task Success Rate (TSR) %**| 35.6     | 76.8     | 79.3     | **81.2** | 67.2       | 61.2             | 55.2             | 64.6        |
> >
> >
> > Our final model, $\textbf{LAM}^4$, consistently outperforms all baselines across key metrics. It achieves a task success rate of 81.2%, compared to 67.2% for GPT-4o, 55.2% for DeepSeek R1, and 64.6% for o1-mini. Similar improvements are seen in step success rate (86.2% vs. 68.8%), object accuracy (87.8% vs. 73.2%), and status accuracy (99.0% vs.  65.3%).
> > These results demonstrate the effectiveness of our domain-specific training pipeline, and show that even a smaller, specialized model like LAM can outperform much larger general-purpose models on practical, real-world tasks. The updated results and comparisons have been integrated into the main text for clarity. **This discussion has been updated in Section 5.1.3.**
> >
> > **b. Justification for Step Precision**
> >
> > We thank the reviewer for raising this point. While task success rate (TSR) is our primary metric, we include step success rate (SSR) as a complementary diagnostic tool. In GUI-based environments like Word, most tasks follow highly constrained and canonical trajectories, making step-level precision both meaningful and necessary. SSR helps reveal whether the agent executes tasks through accurate and reliable plans rather than through brittle or ad-hoc strategies.
> > Although multiple trajectories may theoretically complete a task, in practice, deviations often lead to errors or suboptimal behavior. Thus, LAM’s higher SSR reflects not just closer alignment with reference trajectories but greater consistency, planning quality, and robustness.
> > Also, SSR aligns with standard agent benchmarks (e.g., Mind2Web, AITW). Including SSR ensures a fuller understanding of model behavior beyond binary task outcomes. **We has added the above discussion in the Section 5.1.2 with blue text.**

---

> > > ### Author Response · Authors · 2025-05-30
> > >
> > > **c. Thank you for the comments. We have corrected the typo in Section 4.4 ("where we first The reward model...") with blue text.**
> > >
> > > **d. Would be useful to give a one-line summary of how ReAct works to generate successful trajectories (Section 4.3)**
> > >
> > > To generate meaningful task trajectories for a GUI agent using the ReAct (Reasoning and Acting) mechanism, the agent follows an interleaved Observation–Reasoning–Action cycle for decision making. At each step, the agent perceives the current GUI state, reasons about the next action, executes it, and observes the resulting changes in the environment. This process systematically captures the environment state (including UI trees and screenshots), the action taken, and the agent’s reasoning trace. By aggregating these step-by-step interactions, the system constructs coherent task trajectories that can be used to train a Large Action Model (LAM). **We explain this process in Section 4.3 with blue text.**

---

> > > > ### Comment · Reviewer_CeVy · 2025-06-03
> > > > **Thanks for the revision**
> > > >
> > > > **Difference train and test**.
> > > > Thanks for this, this pipeline makes sense. However, there are a few issues with the revisions. Firstly, you copied your response from the review to the manuscript, but you shouldn't refer to the reviewer in the updated manuscript. Secondly, from the text that is now added to the manuscript the reader cannot verify this pipeline is actually done / how it has been done. What did you use for semantic deduplication? How many datapoints did you throw out using this pipeline? Can you give an example of a training task and a test task and put it in the appendix? What is the standard deviation of the Rouge-L scores? Finally, you don't properly cite "Self-Instruct" in the manuscript.
> > > >
> > > > **Human evaluation**.
> > > > This is really great! But again I'd like to see much more details on this study in the manuscript, as a reader cannot judge it from only the blue paragraph in 7.4. Where are the quantitive results from this study? What exactly did you ask the humans? Will you upload the raw results somewhere?
> > > >
> > > > **GPT-4o tradeoff**.
> > > > This discussion makes sense, thanks for adding it. In the revision, there is a space missing and the writing is a bit off (e.g. the use of the word "actually", consider putting this, and more parts of the manuscript, through chatgpt for revision).
> > > >
> > > > **Shortening the paper**.
> > > > Thanks!
> > > >
> > > > **Baselines**.
> > > > Thanks, though I agree with reviewer yWy8's weakness 2 that agent-style baselines should also be used.
> > > >
> > > > **Writing**.
> > > > - Citation format is still not correct. E.g. revision blue text in section 2. There's no brackets around the citations where there should be (so you're using citet where you should use citep or vice versa)
> > > > - Figure 1 still has incorrect use of English, e.g. "Buy a shoes on the website", "filling the form with the Excel".
> > > > - Many of the blue revision text in the manuscript has typos and incorrect English, e.g. in Section 1, you start with a capital T after a comma where that shouldn't happen.
> > > >
> > > > In general, I'd suggest going over the whole manuscript with an LLM to correct it.

---

> ### Author Response · Authors · 2025-06-04
>
> We thank the reviewer for the second round of detailed and constructive feedback. We appreciate the continued engagement and have revised the manuscript accordingly to address the concerns raised.
>
> **Q1: Difference train and test.**
> We thank the reviewer for the insightful feedback and acknowledge the issues raised. Below, we address each concern in detail and clarify the changes made in the manuscript:
> 1. Rewriting Reviewer Mentions: We have removed any reference to the reviewer in the main paper. The revisions are now presented as part of the method and evaluation pipeline, ensuring the manuscript stands independently and professionally.
> 2. Data Filtering and Deduplication:
>   - We began with 76,851 raw tasks samples. After applying heuristic filters and semantic deduplication (using Sentence-BERT all-MiniLM-L6-v2 with a cosine similarity threshold of 0.95), we removed 179 samples, resulting in 76,672 cleaned tasks.
>   - The final split used for training and evaluation is 1,757 training tasks and 435 test tasks.
>   - We have clarified this in Appendix F with blue text of the revised manuscript.
> 3. ROUGE-L Statistics for Train-Test Separation:
>   - We compute ROUGE-L scores between each test task and its most similar training task.
>   - Mean: 0.27, Std: 0.13, 25th percentile: 0.19, 75th percentile: 0.34.
>   - This ROUGE-L scores distribution is now visualized in the manuscript (Figure 13 in Appendix F).
> 4. Examples of Train-Test Task Pairs:
>  To further demonstrate the dissimilarity between training and test samples, we include three representative examples below, now also added **six examples to Appendix F in the paper:**
>
> PAIR 1 – ROUGE-L Score: 0.2000
> - Test Task: “Attach a digital signature in Word by navigating to the 'Insert' tab, selecting the 'Signature Line' option, and following the prompts to insert and sign the signature line.”
> - Train Task: “Select the 'text to edit' and attempt to access font settings to scale the text within the limitations of the available controls.”
>
> ---
> PAIR 2 – ROUGE-L Score: 0.1250
> - Test Task: “Save a Word document and open the font dialog box for more options.”
> - Train Task: “Navigate to the 'Layout' tab in Word to begin the process of removing page numbers from a specific section.”
>
>
> 5. Citation Fix:
>  We have corrected the reference and inline citation for Self-Instruct (Wang et al., 2022), now properly formatted and included in the bibliography.
>
>
> **Q2: Human evaluation.**
> We thank the reviewer for their appreciation and valuable suggestions regarding the human evaluation. In response, we have significantly expanded the description of our human study in Section 7.4 and added **full quantitative results in Appendix G.**
>
> Our updated content includes:
> - Evaluation Setup: We recruited three expert annotators with prior experience in evaluating LLM-generated content. Each annotator independently rated outputs from LAM across 100 randomly sampled tasks from the test set.
> - Evaluation Criteria: Annotators were asked to rate each output with the question: ``Does the action sequence complete the intended task as described, given the initial application state?"
> - Quantitative Results: We report mean and standard deviation across all ratings in Appendix G.
> - Results Upload: Due to internal policy requirements, the raw human evaluation data must first undergo an internal review process to ensure compliance with data governance and annotator consent protocols. We plan to release the data after this review is complete. The release will include the task descriptions, the initial GUI state for each task (screenshots and XML structure), model-generated action sequences, and corresponding human ratings across evaluation axes. We will provide these materials as supplementary data upon approval, to ensure full transparency and reproducibility.
> We believe these additions provide the necessary transparency and rigor for readers to assess the validity of our human evaluation study.
>
> **Q3: GPT-4o tradeoff.**
> Thank you for the helpful suggestion. We’ve revised the GPT-4o tradeoff discussion to improve clarity and flow, and corrected the formatting issues. We have also proofread the manuscript more thoroughly, including using language refinement tools, to enhance overall readability.

---

> > ### Author Response · Authors · 2025-06-04
> >
> > Q4: Baselines.
> > We appreciate the reviewer’s suggestion and acknowledge the value of comparing with more agent-style baselines. Our current choices were guided by practical and methodological constraints:
> > 1. Offline Benchmark Incompatibility:
> >  Our offline setting evaluates symbolic control-grounded decisions (e.g., click("OK")), allowing precise alignment with annotated ground truth. Agent-style baselines like Gorilla and ScreenAgent generate raw pixel coordinates or bounding boxes, which cannot be reliably mapped to our format without significant vision-grounding extensions, making direct comparison infeasible.
> > 2. ReAct-style Agent Included in Online Evaluation:
> >  Our online setting **(Table 3)** already includes GPT-4o + UFO, which follows a ReAct-style Observation–Reasoning–Action loop. This setup fairly represents the ReAct family in a realistic agent context.
> > 3. Scope and Contributions:
> >  This work focuses on showing that LAM, trained under GPT-4o supervision, can surpass it in both offline and online task execution. Demonstrating that a domain-adapted LAM can outperform its teacher highlights its robustness and practicality.
> >
> > That said, we agree that including more agent-style baselines (e.g., visual-only or Vicuna-based agents) is a valuable direction. We are extending our live benchmark framework to better support such comparisons in future work.
> >
> >
> > Q5: Writing.
> > We thank the reviewer for the careful reading and constructive feedback on the writing and formatting. We have addressed the issues as follows:
> > - Citation Format: We reviewed and corrected all citation commands in the manuscript. Specifically, we replaced improper uses of \cite with \citep to ensure appropriate citation formatting throughout.
> > - Figure 1 Language: We corrected the English descriptions in Figure 1 to ensure grammatical accuracy and clarity.
> > - Typos and Grammar in Revision Text: We carefully edited all newly added content (blue text) for grammatical correctness and consistency with academic writing standards. For instance, the capitalization error after a comma in Section 1 has been corrected. We also performed a full proofreading pass on the manuscript to eliminate remaining typographical and stylistic errors.
> >
> > We appreciate the reviewer’s attention to detail and believe these improvements have significantly enhanced the readability and polish of the paper. Thank you again for your thoughtful and helpful suggestions.

---

> > > ### Comment · Reviewer_CeVy · 2025-06-05
> > > **Thanks for the responses**
> > >
> > > Q1: Difference train and test.
> > > - can you put in the main text in 5.1.1 a reference to appendix F? Also, Appendix F has no mentioning of "using Sentence-BERT all-MiniLM-L6-v2 with a cosine similarity threshold of 0.95", could you add that there too?
> > >
> > > Q2: Human evaluation.
> > > - similarly, put a reference to the relevant appendix section in 7.4 in the main text
> > >
> > > Q3: GPT-4o tradeoff, Q4: Baselines., and Q5: writing
> > > Thanks, these points are adequately addressed in my opinion.

---

> > > > ### Author Response · Authors · 2025-06-06
> > > >
> > > > Thank you for your thoughtful and detailed suggestions. They have significantly helped us improve the clarity, completeness, and overall quality of our manuscript.
> > > >
> > > > **Response to Q1: Difference between Train and Test**
> > > >
> > > > Thank you for highlighting the need for clearer cross-referencing. In the revision, we have added a reference in Section 5.1.1 pointing to Appendix F, which provides additional implementation details for our dataset split methodology. We have also updated Appendix F to explicitly mention that we used Sentence-BERT all-MiniLM-L6-v2 with a cosine similarity threshold of 0.95 for semantic deduplication. This ensures transparency in how semantic similarity was enforced to differentiate training and test samples.
> > > >
> > > > **Response to Q2: Human Evaluation**
> > > >
> > > > We appreciate the suggestion to improve accessibility to human evaluation details. In the revised version, we have added a reference in Section 7.4 directing readers to Appendix G, where we report the quantitative results, evaluation setup, and inter-annotator agreement scores. This helps readers validate the credibility and structure of our human study.
> > > >
> > > > We truly appreciate your insightful feedback! Your guidance has been a beacon throughout our revision process and has greatly elevated the quality of our work.

---

> > > > > ### Comment · Reviewer_CeVy · 2025-06-06
> > > > >
> > > > > Thanks for bearing with me!

---

### Review · Reviewer_KDSj · 2025-03-24

**Summary Of Contributions:**

This paper provides an end-to-end recipe for training a Large Action Model (LAM) from scratch, using GPT4 as an helper tool for building and cleaning data, as well as expert teacher for initial phases of training.

In the proposed scenario, the LAM is implemented on a Windows OS environment, and the type of tasks are software manipulation (opening and editing a document, etc).

The entire procedure is divided into 3 main parts:
Data collection
Training
Evaluation

**Audience:**

Yes

**Broader Impact Concerns:**

As explained in the paper, in 8.2, LAM bring some ethical concerns, which are sufficiently stated in my opinion.

**Claims And Evidence:**

Yes

**Requested Changes:**

My suggestions of changes might depend on the answer to the questions asked above.

**Strengths And Weaknesses:**

*Strengths*

The whole pipeline is very concise and well detailed, clear to read and understand.
Such detailed instruction to train an LAM (or even other large models) are quite rare and very useful.

*Weakness*

I see two main points that remains unclear and limiting the impact of that paper.

1) Is that method applicable to other type of tasks (robotics, medicine, video games etc)? Or are some fundamental modifications required depending on the task? (for ex in robotics, actions are no longer sequences of discrete tokens but take continuous values)

2) Is the proposed method (and especially the training phases) a generic one applied in many papers (and if yes, it could be nice to remind them at the beginning of each section), or is it a specific blend introduced by the present paper (in that case, the paper could be strongly improved by more justifications of the approach, or a comparison with other existing approaches).

Details and Questions:

P6, in 2.1 → what do you mean by physical? the example tasks explored in this paper does not seem physical.

Table 3 --> how is it possible that LAM^2 (from phase 2) outperforms GPT4, since it is basically distilling it in phase 2? also GPT4 does not seems very better than GPT4-mini, maybe something is wrong with GPT4's column?

---

> ### Author Response · Authors · 2025-05-30
>
> We sincerely thank the reviewer for the thoughtful feedback and appreciation of the clarity and usefulness of our pipeline.
>
> **1. Applicability to Other Domains**
>
> We thank the reviewer for raising this important point. Our pipeline is fundamentally domain-agnostic, it consists of: (1) Data Collection & Task‐Plan Pretraining (2) Imitation Learning from Expert Trajectories (3) Self-Boosting Exploration (4) Reward-Model Guided Refinement
>
> However, to deploy in a new domain you must satisfy: It is able to execute, observe, and validate action trajectories (e.g., via APIs, simulators, or instrumentation).
>
> With those in place, the same four-phase structure applies: (1) Video Games: discrete controls (buttons, joystick directions) can be collected via game APIs or screen‐state instrumentation. (2) Robotics: the same stages apply but the model must predict continuous motor commands or trajectories (e.g. via a different decoder head). Simulation platforms (e.g., MuJoCo, Gazebo) make data collection low-risk and scalable. (3) Web & OS Automation: as demonstrated, discrete GUI events work out of the box.
> In contrast, high-stakes domains and low tolerance for errors like medicine: one cannot freely execute and log actions on real patients. In such cases, careful sandboxing (e.g., digital twins) or expert‐in-the-loop validation would be required before deployment. We will clarify these domain requirements and potential adaptations in the revised manuscript.
>
>
> **We have added this discussion in Section 4.4.2 with blue text.**
>
> **2. Novelty of Training Phases**
>
> Our key contribution is propose the full action modeling pipeline (including data collection, grounding, and evaluation) which is developed from scratch for this specific setting. Unlike prior works that assume access to curated trajectories or operate in simulators, our pipeline:
> - Bootstraps all data—task plans and action trajectories—via real environment interaction, with no pre-existing task-action pairs.
> - Interleaves training and data collection, enabling self-improvement through failure harvesting.
> - Introduces novel modules such as reward-guided filtering and grounding evaluation.
> While individual components may resemble prior methods, their integration into a self-bootstrapped, adaptive pipeline for LAM development is, to our knowledge, unique. **We have emphasize this in the revised paper and cite relevant baselines for context in Section 4 with blue text.**
>
> **3.  Detailed Questions & Corrections**
>
> **a. “Physical” vs. “Digital” Interactions **
>
> We apologize for the ambiguity. In this paper, “physical” refers broadly to any non-text output of an agent—here, GUI interactions on a desktop OS. We will revise to distinguish (1) Digital actions: GUI clicks, API calls (2) Physical actions: e.g. robotic motor commands.
>
> **b. Table 3: LAM² vs. GPT-4o Performance**
>
> - Why LAM² > GPT-4o?
>   We appreciate the reviewer’s observation and would like to clarify a potential misunderstanding. While LAM² is fine-tuned on trajectories labeled by GPT-4o, it is not a simple distillation of GPT-4o’s capabilities. Rather, it builds upon a broader foundation:
>   - LAM¹ first undergoes task-plan pretraining, learning structured decomposition and execution logic from curated plans. This planning knowledge is not explicitly present in GPT-4o’s responses and provides a strong prior that LAM² benefits from.
>   - GPT-4o is used only for labeling, not as the exclusive source of reasoning or behavior. Thus, LAM² combines structured planning priors (from LAM¹) with high-quality action labels, resulting in more grounded and consistent execution.
>   - Moreover, GPT-4o remains a zero-shot model, susceptible to hallucinating non-existent UI elements or inconsistent behaviors in unfamiliar environments. In contrast, LAM² is domain-adapted and fine-tuned to the task interface, improving both accuracy and reliability.
>  **We have clarified this distinction in the revised paper (Section 5.1.3).**
>
> - GPT-4o vs. GPT-4o-Mini anomalies:
>   Thank you for catching this inconsistency. Upon review, we identified an error in the evaluation of GPT-4o-Mini due to an incorrect denominator—responses with format violations were mistakenly excluded from the calculation. This led to inflated performance metrics.
>  After correcting the evaluation script, GPT-4o now consistently outperforms GPT-4o-Mini across all step-level metrics, as expected. **We have updated Table 3 (now Table 2) with the corrected scores and clarified this change in the revised manuscript (Section 4.1.3).**

---

> > ### Author Response · Authors · 2025-05-30
> >
> > **4. Broader Impact**
> > We appreciate the reviewer’s feedback and will expand Section 8.1 with blue text ("Mitigation Strategies for Safe Deployment") with concrete mitigation strategies. Specifically, we will employ sandboxed environments, human-in-the-loop verification, and action validation to ensure safe execution. We also could implement progressive deployment to minimize risk and support responsible LAM operation in real-world environments.
> >
> > **We added a new subsection in Section 8.1 "Mitigation Strategies for Safe Deployment" with detailed ethical considerations.**

---

### Review · Reviewer_yWy8 · 2025-05-12

**Summary Of Contributions:**

This paper presents Large Action Models (LAMs), a framework for building language-based agents capable of controlling desktop GUI environments. The authors propose a practical, reproducible workflow for training and evaluating action agents, including synthetic data generation, imitation learning, self-bootstrapping, and offline reinforcement learning. They develop a two-stage, LLM-driven pipeline to automatically produce over 2,000 verified task-action trajectories with minimal human labeling. The detailed comparisons are provided with GPT-4-o, demonstrating that a text-only LAM achieves comparable or better performance at lower execution latency.

**Audience:**

Yes

**Broader Impact Concerns:**

The Broader Impact Statement should address these concerns by outlining mitigation strategies, such as ensuring transparent decision-making, incorporating privacy safeguards, actively monitoring for biases, and discussing equitable access to LAM technology.

**Claims And Evidence:**

No

**Requested Changes:**

(**RC1**) The paper acknowledges failure modes and ethical risks only briefly and lacks detailed analysis of robustness, failure cases, or safety in long-horizon or out-of-distribution tasks.

(**RC2**) Some of the sentences have the risk of over-claiming:

- *"We are the first to propose Large Action Models."* Prior arts, such as RT-2, ScreenAgent, and OS-Copilot, use different names but the same idea. This work sounds more like extend the concept of large language and vision–language models to desktop GUI control.

- *"Comprehensive pipeline not shown before."* Similar multi-stage pipelines (plan / act / reflect / RL) exist in quite a few agentic workflow papers.

(**RC3**) The work claims that their pipeline can scales without expensive human labels. However it still depends heavily on GPT-4-o for both trajectory validation and task augmentation, which involves significant cost and circularity. The authors need to provide cost analysis and discuss how cheaper open-source models could replace GPT-4o in the loop.

(**RC4**) The paper uses inconsistent citation formats.

**Strengths And Weaknesses:**

(**S1**) Unlike papers limited to simulations or synthetic web interfaces, this work is deployed in a live Windows environment (Microsoft Word), enhancing its practical relevance.

(**S2**) The paper delivers a complete pipeline, from synthetic data generation to live deployment.

(**W1**) The notion of a “Large Action Model” is largely a rebranding of prior work in action-based LLM agents (e.g., OS-Copilot, RT-2). The conceptual framing is not as original as claimed.

(**W2**) Comparisons are only made to GPT-4o. Other relevant strong baselines like ReAct-style agents, open-source GUI agents (e.g., Gorilla, ScreenAgent), or fine-tuned Vicuna models are omitted.

---

> ### Author Response · Authors · 2025-05-30
>
> We sincerely thank the reviewer for the thoughtful feedback and appreciation of the clarity and usefulness of our pipeline.
>
> **RC1: Robustness, Failure Modes & Safety Analysis**
>
> We thank the reviewer for highlighting the need for a deeper analysis of model robustness and failure modes. In response, we have added a dedicated section on Failure Cases and Error Patterns in the revised manuscript (Section 5.2 with blue text).
> These failure cases reveal three dominant error patterns in LAM’s current behavior: (1) premature task termination due to overconfident stopping, (2) insufficient attention to task preconditions, and (3) UI disambiguation failures when interacting with similar elements. Addressing these issues will require improvements in planning, state verification, and more robust grounding to UI context, which will be considered in out future work.
>
> **RC2: Claim Refinement & Prior Art Acknowledgment**
>
>  We appreciate the reminder to appropriately situate our contributions. In the updated paper, we will:
> - We will clarify that our novelty lies in a unified, end-to-end pipeline, from zero human labels to live desktop GUI deployment.
> - Moderate phrasing around “first to propose” and “comprehensive pipeline,” replacing them with “to the best of our knowledge” and “a streamlined, unified workflow,” while citing related multi-stage agent papers. **We have add these citations in the manuscript in Section 4.2, and 4.4.1.**
>
> **RC3: Cost Analysis & Open-Source Alternatives**
>
> We agree that reliance on GPT-4o introduces cost and circularity concerns. To address this, **we include a cost analysis in the revised manuscript (Section 4.1.3):** GPT-4o was used for approximately 50K API calls during data instantiation and validation, incurring a total cost of $~2.5K$ (at $0.05$ per 1K tokens), compared to an estimated $>20K$ for equivalent manual annotation. Inference costs are minimal ($<0.01$ per task).
>
> **RC4: Citation Formatting Consistency**
>
> Thank you for pointing this out. We have standardized all references to the “Author et al. (Year)” format and ensure uniform bibliography styling throughout the paper.
>
> **Broader Impact**
> We appreciate the reviewer’s feedback and will expand Section 8.1 with blue text ("Mitigation Strategies for Safe Deployment") with concrete mitigation strategies. Specifically, we will employ sandboxed environments, human-in-the-loop verification, and action validation to ensure safe execution. We also could implement progressive deployment to minimize risk and support responsible LAM operation in real-world environments.
>
> **We added a new subsection in Section 8.1 "Mitigation Strategies for Safe Deployment" with detailed ethical considerations.**

---

### Decision · Action_Editor_kmxj · 2025-06-27

**Recommendation:** Accept with minor revision

**Audience:**

Yes

**Audience Explanation:**

Agentic views of LLMs are a popular topic, and I think many interested in this broad space will enjoy reading the paper and learning about the real-world experimental setting for models performing computer control actions on a real OS.

**Claims And Evidence:**

Yes

**Claims Explanation:**

While there were some perhaps overclaimed contributions of the paper (see reviewer comments and requested changes), the majority of reviewers were in favour of publishing this work showcasing OS control from large action models (effectively LLMs with specific training towards UI interaction). Reviewers appreciated the detailed recipe for producing similar models, the real world setting of acting on Windows, and generally found the experiments to be convincing (with one reviewer wishing for a broader base of experiments). In my opinion this paper meets the acceptable standard for publication on these grounds.

---

> ### Author Response · Authors · 2025-07-11
>
> Dear Reviewers and Action Editor,
>
> Thank you very much for your thoughtful and constructive feedback throughout the review process. We truly appreciate your recognition of our work, particularly the value of demonstrating OS-level control using large action models in a real-world environment. We're encouraged by your comments regarding the broader interest this topic may have within the TMLR community.
>
> We would also like to kindly inform you that we have uploaded the final camera-ready version of the paper, incorporating the suggested revisions. Thank you again for your time and support in bringing this work to publication.
>
> Warm regards,
> The Authors